# Out-of-Distribution Detection with Relative Angles

**Berker Demirel**
Institute of Science and Technology Austria
3400 Klosterneuburg, Austria
`berker.demirel@ist.ac.at`

**Marco Fumero**
Institute of Science and Technology Austria
3400 Klosterneuburg, Austria
`marco.fumero@ist.ac.at`

**Francesco Locatello**
Institute of Science and Technology Austria
3400 Klosterneuburg, Austria
`francesco.locatello@ist.ac.at`

## Abstract

Deep learning systems deployed in real-world applications often encounter data that is different from their in-distribution (ID). A reliable model should ideally abstain from making decisions in this out-of-distribution (OOD) setting. Existing state-of-the-art methods primarily focus on feature distances, such as k-th nearest neighbors and distances to decision boundaries, either overlooking or ineffectively using in-distribution statistics. In this work, we propose a novel angle-based metric for OOD detection that is computed relative to the in-distribution structure. We demonstrate that the angles between feature representations and decision boundaries, viewed from the mean of in-distribution features, serve as an effective discriminative factor between ID and OOD data. We evaluate our method on nine ImageNet-pretrained models. Our approach achieves the lowest FPR in 5 out of 9 ImageNet models, obtains the best average FPR overall, and consistently ranking among the top 3 across all evaluated models. Furthermore, we highlight the benefits of contrastive representations by showing strong performance with ResNet SCL and CLIP architectures. Finally, we demonstrate that the scale-invariant nature of our score enables an ensemble strategy via simple score summation. Code is available at https://github.com/berkerdemirel/ORA-OOD-Detection-with-Relative-Angles.

## 1 Introduction

A trustworthy deep learning system should not only produce accurate predictions, but also recognize when it is processing an unknown sample. The ability to identify when a sample deviates from the expected distribution, and potentially rejecting it, plays a crucial role especially in safety-critical applications, such as medical diagnosis [Fernando et al., 2021], driverless cars [Bogdoll et al., 2022] and surveillance systems [Diehl and Hampshire, 2002]. The out-of-distribution (OOD) detection problem addresses the challenge of distinguishing between in-distribution (ID) and OOD data – essentially, drawing a line between what the system knows and what it does not.

Various approaches have been proposed for OOD detection, mainly falling into two categories: (i) methods that suggest model regularization during training [Lee et al., 2018, Hendrycks et al., 2019, Meinke and Hein, 2020, Liu and Qin, 2025, Ammar et al., 2024], and (ii) post-hoc methods, which leverage a pre-trained model to determine if a sample is OOD by designing appropriate *score functions* [Peng et al., 2024, Hendrycks and Gimpel, 2022, Sun et al., 2022]. Post-hoc methods are more advantageous for their efficiency and flexibility, as they can be applied to arbitrary pre-trained models without retraining. These approaches are often categorized based on the domain of their score functions, i.e., at which representational abstraction level they assess if a sample is OOD or

not. Earlier techniques focus on measuring the model confidence in the logits space [Hendrycks and Gimpel, 2022, Liu et al., 2020], while the recent works employ distance-based scores [Sun et al., 2022, Sehwag et al., 2021] defined in the model feature space. While logit-based methods suffer from the overconfident predictions of neural networks [Minderer et al., 2021, Lakshminarayanan et al., 2017, Guo et al., 2017], the recent success of distance-based techniques highlights that the relationships in the latent space can provide a richer analysis.

A natural approach to feature representations is by checking their proximity to the decision boundaries [Liu and Qin, 2024]. Conceptually, this can be related to identifying hard-to-learn examples in data-efficient learning [Joshi et al., 2024, Chen et al., 2023]. OOD samples can be viewed as hard-to-learn since they do not share the same label distribution as ID data. The success of this approach has been directly showed in fDBD score from Liu and Qin [2024]. However, our derivations revealed that the regularization term they use to incorporate ID statistics introduces an additional term that does not correlate with ID/OOD separation, ultimately hindering their performance.

In this work, we present OOD Detection with Relative Angles (ORA), a novel approach that exploits the relationship between feature representations and classifier decision boundaries, in the context of the mean statistics of ID features. Unlike the earlier techniques, ORA introduces a new angle-based measure that calculates the angles between the feature representations and their projection onto the decision boundaries, relative to the the mean statistics of ID features. Changing reference frame to the mean of ID features adds another layer of discriminatory information to the score, as it naturally incorporates the ID statistics to the distance notion, exploiting the disparity between ID and OOD statistics. Moreover, the scale-invariant nature of angle-based representations, as similarly observed in Moschella et al. [2023], allows us to aggregate the confidence scores from multiple pre-trained models simply by summing their ORA scores. This enables to have a score that can be single model based or extended to ensemble of models. In summary, our key contributions include:

- We present a novel post-hoc OOD score, which computes the angles between the feature representation and its projection to the decision boundaries, relative to the mean of ID-features

- We conduct an extensive evaluation on the ImageNet OOD benchmark using 9 model backbones, including modern transformer-based architectures and compare 12 detection methods. ORA achieves the best average FPR95 across all models, ranks in the top 3 for every model, and is the best-performing method on 5 out of 9 models.

- We analyze the benefits of using contrastively learned features with ORA. Our method achieves the best performance on CIFAR-10 with ResNet18-SCL and on ImageNet with ResNet50-SCL, reducing average FPR95 by $0.88\%$ and $7.74\%$ respectively. We further validate this trend with the CLIP model, where ORA achieves strong results in both zero-shot ($25.85\%$ FPR95) and linear probing ($23.94\%$ FPR95) settings.

- ORA's scale-invariance allows aggregation of confidence scores from multiple pre-trained models, enabling an effective ensemble. Our experiments show that the ORA ensemble reduces the FPR95 by $2.51\%$ on the ImageNet OOD benchmark compared to the best single model performance.

## 2   Related Work

Previous work in OOD detection falls into two categories: (i) methods that regularize models during training to produce different outcomes for ID and OOD data, and (ii) post-hoc methods that develop scoring mechanisms using pre-trained models on ID data.

**Model regularization.**   Early methods addressing the OOD detection problem Bevandić et al. [2019], Hendrycks et al. [2019] utilize additional datasets to represent out-of-distribution data, training models with both positive and negative samples. This approach assumes a specific nature of OOD data, potentially limiting its effectiveness when encountering OOD samples that deviate from this assumption during inference. Malinin and Gales [2018] designed a network architecture to measure distributional uncertainty. In Geifman and El-Yaniv [2019]'s work, they provide another architecture with an additional reject option to abstain from answering. Ming et al. [2022], Du et al. [2022], focused on synthesizing outliers rather than relying on auxilary datasets. On the other hand, Van Amersfoort et al. [2020], Wei et al. [2022] argued that overconfident predictions of the networks on OOD data are the problem to be mitigated. For example, Van Amersfoort et al. [2020] puts an additional gradient

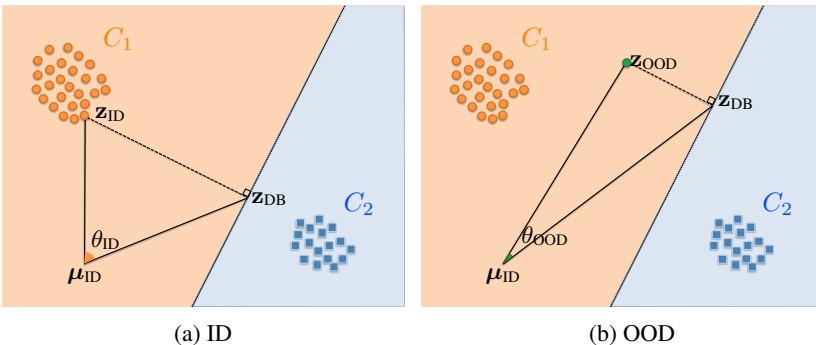

(a) ID             (b) OOD

Figure 1: Geometric visualization of ORA for in-distribution (*left*) and out-of-distribution (*right*) cases. ORA focuses on the angular distance between the feature representation and the decision boundary, from the perspective of the in-distribution mean. The angle $\theta$ serves as the distinguishing factor between ID and OOD samples, with $\theta_{\text{ID}} > \theta_{\text{OOD}}$.

penalty to limit the confidence of the network. Whereas, Wei et al. [2022] tackled the same problem by enforcing a constant logit vector norm during training. Although it is natural to impose structures during training for better OOD detection, these methods face the trade-off between OOD separability and model performance. Moreover, such approaches lack the flexibility of post-hoc score functions, as they necessitate model retraining which can be both time-consuming and computationally expensive.

**Score functions.** Recently, developing score functions for pretrained models on ID data has gained attention due to its ease of implementation and flexibility. These methods typically either couple feature representations with distance metrics, or measure a model's confidence using its logits. Beyond canonical works such as Maximum Softmax Probability [Hendrycks and Gimpel, 2022], ODIN score [Liang et al., 2018], Energy score [Liu et al., 2020], we observed many advancements in post-hoc score design. For example, the activation shaping algorithms such as ASH [Djurisic et al., 2023], Scale [Xu et al., 2024b], and ReAct [Sun et al., 2021], apply activation truncations to feature representations, reducing model's confidence for OOD data. These methods can be used in conjunction with ORA improving the performance. Recent distance-based methods KNN [Sun et al., 2022], NNGuide [Park et al., 2023], R-Mah [Ren et al., 2021], Maha+ [Müller and Hein] and fDBD [Liu and Qin, 2024] successfully utilized the feature representations from networks. Both NNGuide and KNN assign a score to a sample based on the kth nearest neighbor in ID training set. R-Mah measures the relative Mahalanobis distance between class centroids and the overall data centroid. In contrast, fDBD assigns a score to a sample based on its estimate of the distance between the feature representation and the decision boundaries. Moreover, recent angle-based methods such as P-Cos [Galil et al.] and R-Cos [Bitterwolf et al., 2023] assign scores based on the cosine similarity between representations and class centroids.

Our work falls into the score function category, serving as a plug-in for any pre-trained model on ID data. ORA combines feature space and logit space methods by utilizing the relative angle between the feature representation and its projections to the decision boundaries. Among the existing works, the closest approach to our method is fDBD, which uses a lower bound estimate to the decision boundaries. However, the regularization term they introduced inadvertently includes a term in their equation that is uncorrelated with being OOD or ID and can change spuriously, impeding performance. In contrast, we provide a score function that effectively incorporates in-distribution context and maintains scale invariance, all without extra regularization terms.

## 3 Method

**Problem Setting.** We consider a supervised classification setting with input space $\mathcal{X}$ and label space $\mathcal{Y}$, following the literature Yang et al. [2024]. Given a model $f : \mathcal{X} \to \mathbb{R}^{|\mathcal{Y}|}$ pretrained on an in-distribution dataset $D_{\text{ID}} = \{(\mathbf{x}_i, y_i)\}_{i=1}^{N}$, where elements of $D_{\text{ID}}$ are drawn from a joint distribution $P_{\mathcal{X}\mathcal{Y}}$, with support $\mathcal{X} \times \mathcal{Y}$. We denote its marginal distribution on $\mathcal{X}$ as $P_{\text{ID}}$. The *OOD detection problem* aims to determine whether an input sample originates from the in-distribution $P_{\text{ID}}$ or not. Let $\mathcal{Y}_{\text{OOD}}$ be a set of labels such that $\mathcal{Y} \cap \mathcal{Y}_{\text{OOD}} = \emptyset$. OOD samples are drawn from a distribution $P_{\text{OOD}}$ which is the marginal distribution on $\mathcal{X}$ of the joint distribution over $\mathcal{X} \times \mathcal{Y}_{\text{OOD}}$

i.e., they share the same input space $\mathcal{X}$ as in-distribution samples, but have labels outside $\mathcal{Y}$. Shifting from $P_{\text{ID}}$ to $P_{\text{OOD}}$ corresponds to a semantic change in the label space.

The OOD decision can be made via the function $d : \mathcal{X} \to \{\text{ID}, \text{OOD}\}$ given a *score function* $s : \mathcal{X} \to \mathbb{R}$ such that:

$$d(\mathbf{x}; s, f) = \begin{cases} \text{ID} & \text{if } s(\mathbf{x}; f) \geq \lambda \\ \text{OOD} & \text{if } s(\mathbf{x}; f) < \lambda \end{cases}$$

where samples with high scores are classified as ID, according to the threshold $\lambda$. For example, to compute the standard FPR95 metric [Yang et al., 2024], the threshold $\lambda$ is chosen such that it correctly classifies 95% of ID held-out data. An ideal OOD score function should capture differences in model outputs between samples drawn from $P_{\text{ID}}$ and $P_{\text{OOD}}$, effectively detecting inputs from unseen classes.

### 3.1 Out-of-Distribution Detection with Relative Angles

This section introduces our OOD score function, which leverages relative angles in feature space to relate feature representations to decision boundaries, distinguishing ID from OOD samples. Figure 1 provides a geometric visualization of our method. Our approach leverages the geometric relationships between three key points in the feature space: (i) the initial representation of a sample, (ii) its projection onto the decision boundary, and (iii) the mean of in-distribution features.

We propose using the relation between feature representations and decision boundaries by deriving closed-form plane equations for the decision boundaries between any two classes. Specifically, we examine the angle formed between the feature representation vector and its projection onto the decision boundary. However, this angle is sensitive to the choice of origin, creating an ambiguity as the geometric relationship between the feature representation and the decision boundary should be translation-invariant. To address this, we propose to represent features in a reference frame relative to the mean of the in-distribution samples. Therefore, we incorporate ID characteristics by centering around its mean, while ensuring scale and translation invariance.

We observe that the angle between the centered representation and its projection onto the decision boundary is larger for ID data, indicating them requiring higher cost to change their label which captures the model's confidence. In contrast, for OOD data, angle is smaller since they are expected to be more unstable, as they do not contain strong clues about their predicted classes (see Figure 1 for a conceptual explanation).

Our framework provides a concise scoring with useful properties such as translation and scale invariance. These properties enable ORA to be used in conjunction with existing activation shaping algorithms and allow for confidence aggregation across different models through score summation.

### 3.2 Features on the Decision Boundary

In this section, we derive the mathematical equations and demonstrate the properties of our score. The model $f$ can be rewritten as a composed function $f_1 \circ ... f_{L-1} \circ g$, where $L$ is the number of layers and $g : \mathbb{R}^D \to \mathbb{R}^{|\mathcal{Y}|}$ corresponds to the last layer classification head. The function $g(\mathbf{z}) = \mathbf{W}\mathbf{z} + \mathbf{b}$ maps penultimate layer features $\mathbf{z} \in \mathbb{R}^D$ to the logits space via $\mathbf{W} \in \mathbb{R}^{|\mathcal{Y}| \times D}$ and $\mathbf{b} \in \mathbb{R}^{|\mathcal{Y}|}$. The decision boundary between any two classes $y_1$ and $y_2$ with $y_1 \neq y_2$ can be represented as:

$$DB_{y_1, y_2} = (\mathbf{w}_{y_1} - \mathbf{w}_{y_2})^T \mathbf{z} + b_{y_1} - b_{y_2} = 0$$

where $\mathbf{w}_{y_1}$ (or $\mathbf{w}_{y_2}$) denotes the the row vectors of $\mathbf{W}$ corresponding to class $y_1$ (respectively $y_2$) and similarly, $b_{y_1}, b_{y_2}$ are the bias values corresponding to classes $y_1$ and $y_2$. Intuitively, given a fixed classifier, this equation is satisfied for all $\mathbf{z}$'s such that their corresponding logits for class $y_1$ and $y_2$ are equal. Then, feature representations can be projected onto the hyperplane that defines the decision boundary:

$$\mathbf{z}_{db} = \mathbf{z} - \frac{(\mathbf{w}_{y_1} - \mathbf{w}_{y_2})^T \mathbf{z} + (b_{y_1} - b_{y_2})}{\|\mathbf{w}_{y_1} - \mathbf{w}_{y_2}\|^2}(\mathbf{w}_{y_1} - \mathbf{w}_{y_2}) \tag{1}$$

Let $\boldsymbol{\mu}_{\text{ID}} \in \mathbb{R}^D$ be the mean of the in-distribution feature representations. Centering w.r.t. $\boldsymbol{\mu}_{\text{ID}}$ corresponds to shifting the origin to $\mu_{\text{ID}}$. In this new reference frame, three key points form a triangle in $D$-dimensional space: the centered feature vector $(\mathbf{z} - \boldsymbol{\mu}_{\text{ID}})$, its projection onto the decision boundary $(\mathbf{z_{db}} - \boldsymbol{\mu}_{\text{ID}})$ and the new origin (see Figure 1). Then, rather than the absolute distance between $\mathbf{z}$ and $\mathbf{z}_{db}$, we use the relative angle $\theta_{y_1,y_2}(\mathbf{z})$ from the in-distribution feature representation's reference frame. This captures *the position of features and the decision boundaries relative to the in-distribution data*, while also being scale invariant:

$$\theta_{y_1,y_2}(\mathbf{z}) = \arccos\left(\frac{\langle \mathbf{z} - \boldsymbol{\mu}_{\text{ID}}, \mathbf{z}_{db} - \boldsymbol{\mu}_{\text{ID}}\rangle}{\|\mathbf{z} - \boldsymbol{\mu}_{\text{ID}}\| \cdot \|\mathbf{z}_{db} - \boldsymbol{\mu}_{\text{ID}}\|}\right) \tag{2}$$

Our score function captures the maximum discrepancy of the relative angles between the centered feature representation and its projections on $DB_{\hat{y},y'}$, where $\hat{y} = \arg\max_{y \in \mathcal{Y}} g(\mathbf{z})$ is the predicted class, and $y' \in \mathcal{Y}$, $y' \neq y$. Therefore for a sample $\mathbf{x} \in \mathcal{X}$, given $\mathbf{z} = f_1 \circ ... \circ f_{L_1}(\mathbf{x})$ we can write the score $s(\mathbf{x}, f)$ as a function of $\mathbf{z}$:

$$\tilde{s}(\mathbf{z}) = \max\left(\{\theta_{y,y'}(\mathbf{z})\}_{y' \in \mathcal{Y}, y' \neq y}\right) \tag{3}$$

Intuitively, our score function captures several key aspects:

- **Confidence measure.** The angle between the feature representation and its projection onto a decision boundary is proportional to the distance between them, serving as a proxy for the model's confidence.

- **In-distribution context.** By centering the space using the mean of in-distribution features, we incorporate ID statistics, improving angle separability across points.

- **Maximum discrepancy.** It selects the furthest decision boundary by finding the maximum angle across classes. This captures the model's confidence in the least likely class.

- **Scale invariance.** Unlike absolute distances, angles remain consistent even if the feature space is scaled, allowing for fair comparisons between different models. See Appendix J for a theoretical justification of angle-based scores.

**Relation with the state-of-the-art fDBD [Liu and Qin, 2024]).** We now provide a geometric interpretation for the score function fDBD. Using our analysis, we identified that their score can directly be mapped into the triangle we formed in Figure 1. For a sample $\mathbf{x} \in \mathcal{X}$:

$$\text{fDBD}(\mathbf{z}) = \frac{d(\mathbf{z}, \mathbf{z}_{db})}{\|\mathbf{z} - \boldsymbol{\mu}_{\text{ID}}\|}$$

where $\mathbf{z} \in \mathbb{R}^D$ is the feature representations of the input $x \in \mathcal{X}$, $\mathbf{z}_{db} \in \mathbb{R}^D$ is its projection onto the decision boundary, and $d(\cdot, \cdot)$ is the euclidean distance. Although seemingly unrelated, we can connect this score to our relative angle and demonstrate that the regularization term on the denominator brings a term that does not effectively discriminate between OOD and ID. Using translation invariance of the euclidean distance, the same score can be written as:

$$\text{fDBD}(\mathbf{z}) = \frac{d(\mathbf{z} - \boldsymbol{\mu}_{\text{ID}}, \mathbf{z}_{db} - \boldsymbol{\mu}_{\text{ID}})}{d(\mathbf{z} - \boldsymbol{\mu}_{\text{ID}}, 0)}$$

One can observe that, this is the ratio of two sides of the triangle formed between the points $\mathbf{z} - \boldsymbol{\mu}_{\text{ID}}$, $\mathbf{z_{db}} - \boldsymbol{\mu}_{\text{ID}}$ and the origin. Using the law of sines:

$$\frac{d(\mathbf{z} - \boldsymbol{\mu}_{\text{ID}}, \mathbf{z}_{db} - \boldsymbol{\mu}_{\text{ID}})}{\sin(\theta)} = \frac{d(\mathbf{z} - \boldsymbol{\mu}_{\text{ID}}, 0)}{\sin(\alpha)} \Rightarrow \frac{\sin(\theta)}{\sin(\alpha)} = \frac{d(\mathbf{z} - \boldsymbol{\mu}_{\text{ID}}, \mathbf{z}_{db} - \boldsymbol{\mu}_{\text{ID}})}{d(\mathbf{z} - \boldsymbol{\mu}_{\text{ID}}, 0)} = \text{fDBD}(\mathbf{z}) \tag{4}$$

where $\theta$ and $\alpha$ are the angles opposite to the sides $\mathbf{z} - \boldsymbol{\mu}_{\text{ID}} - (\mathbf{z_{db}} - \boldsymbol{\mu}_{\text{ID}})$ and $\mathbf{z} - \boldsymbol{\mu}_{\text{ID}}$ respectively. Although the observation they made on comparing the distances to the decision boundaries at equal

deviation levels from the mean of in-distribution is inspiring, we claim that the angle $\alpha$ is not very informative for ID and OOD separation. This is because $\alpha$ is connected to the magnitude of the feature vector relative to $\boldsymbol{\mu}_{\text{ID}}$, which may not directly correlate with OOD characteristics. On Figure 2 we show the $\sin(\alpha)$ values between CIFAR-10 [Krizhevsky, 2009] and Texture [Cimpoi et al., 2014] datasets, empirically justifying that including this term impedes fDBD's performance. Omitting the denominator from Equation 4 allows to effectively capture the relation between the feature representation and the decision boundary from the mean of in-distribution's view. See Appendix J for more detailed explanation on the performance relationship between raw distance, fDBD and ORA.

## 4 Experiments

In this section, we first test ORA on a large-sale ImageNet OOD benchmark [Deng et al., 2009] that spans nine models –including ConvNeXt [Liu et al., 2022], Swin [Liu et al., 2021], DeiT [Touvron et al., 2021] and EVA [Fang et al., 2023]– to establish the performance beyond the usual ResNet-50 [He et al., 2016] setting. We then demonstrate how ORA benefits from contrastively learned features (i) on CLIP [Radford et al., 2021] in both zero-shot and linear-probe modes, and (ii) on CIFAR-10 [Krizhevsky, 2009] and ImageNet with ResNet18/50 checkpoints trained with supervised contrastive (SCL) loss [Khosla et al., 2020]. Next, we show that ORA's scale-invariant scores can be ensemble-summed across architectures for additional gains, and that it pairs seamlessly with post-hoc activation-shaping methods such as ReAct and ASH. Finally, we present an ablation study to asses the contributions of key design choices.

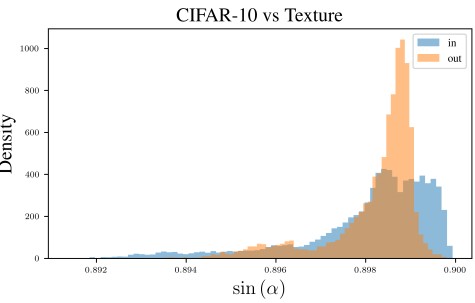

Figure 2: Histogram of ID (CIFAR-10) and OOD (Texture) samples with respect to the sine of the angle formed with the vector $\mathbf{z} - \boldsymbol{\mu}_{\text{ID}}$. This empirically shows that $\sin(\alpha)$ is not highly informative for distinguishing ID from OOD.

**Benchmarks.** We consider two widely used benchmarks: CIFAR-10 [Krizhevsky, 2009] and ImageNet [Deng et al., 2009]). We included the evaluation on CIFAR-10 OOD Benchmark to show the performance on smaller scale datasets. In CIFAR-10 experiments, we use a pretrained ResNet-18 architecture He et al. [2016] trained with supervised contrastive loss [Khosla et al., 2020], following previous literature Liu and Qin [2024], Sun et al. [2022], Sehwag et al. [2021]. During inference 10.000 test samples are used to set the in-distribution scores and choose the threshold value $\lambda$; while the datasets SVHN [Netzer et al., 2011], iSUN [Xu et al., 2015], Places365 [Zhou et al., 2017] and Texture [Cimpoi et al., 2014] are used to obtain out-of-distribution scores and metric evaluation.

For the large-scale ImageNet OOD benchmark, we extend prior evaluations [Liu and Qin, 2024, Sun et al., 2022, Park et al., 2023, Ren et al., 2021, Sun et al., 2021, Xu et al., 2024b] by going beyond ResNet-50 and including a diverse set of nine models, such as ConvNeXt [Liu et al., 2022], Swin Transformer [Liu et al., 2021], DeiT [Touvron et al., 2021], and EVA [Fang et al., 2023], along with their ImageNet-21k pretrained counterparts when available. This broader evaluation allows a more comprehensive assessment of OOD detection performance across modern architectures. A validation set of 50,000 ImageNet samples is used to set ID scores and the threshold, while the OOD datasets include iNaturalist [Van Horn et al., 2018], SUN [Xiao et al., 2010], Places365 [Zhou et al., 2017], and Texture [Cimpoi et al., 2014]. Note that, in this typical OOD Detection Benchmarks the samples that have same classes as ID are removed from their OOD counterparts, following the work Huang and Li [2021] and fitting into our problem setting.

**Metrics.** We report two metrics in our experiments: False-positive rate at $\%95$ true positive rate (FPR95), and Area Under the Receiver Operating Characteristic Curve (AUROC). FPR95 measures what percentage of OOD data we falsely classify as ID where our threshold includes $95\%$ of ID data. Therefore, smaller FPR95 indicates a better performance by sharply controlling the false positive rate. On the other hand, AUROC measures the model's ability to distinguish between ID and OOD by calculating the area under the curve that plots the true positive rate against the false positive rate across thresholds. AUROC shows how rapidly we include ID data while paying the cost of including false positives. Thus, higher AUROC shows a better result.

Table 1: Average FPR95 (%) on the ImageNet OOD detection benchmark. Each column corresponds to a different detection method, and each row to a model backbone. Lower values indicate better performance. The best result per row is highlighted in **bold**.

| Model | MSP | MaxL | Energy | R-Mah | R-Cos | P-Cos | NNG | KNN+ | fDBD | NCI | Maha+ | NECO | ORA |
|---|---|---|---|---|---|---|---|---|---|---|---|---|---|
| ConvNeXt | 63.54 | 60.69 | 78.70 | 45.79 | 46.99 | 49.37 | 39.54 | 47.40 | 39.03 | 36.44 | 36.68 | 34.52 | **34.92** |
| ConvNeXt-pre | 45.57 | 42.39 | 46.95 | 34.91 | 36.16 | 35.47 | 76.91 | 77.23 | 42.13 | 50.48 | 37.50 | 34.40 | **29.53** |
| Swin | 61.99 | 59.21 | 65.78 | **45.25** | 46.26 | 49.37 | 57.34 | 53.79 | 52.34 | 57.25 | 48.21 | 58.24 | 51.56 |
| Swin-pre | 46.20 | 40.86 | 42.33 | 39.40 | 35.66 | **33.94** | 44.06 | 51.34 | 38.92 | 39.63 | 42.67 | 36.49 | 35.13 |
| DeiT | 57.48 | 57.52 | 70.05 | 52.51 | **47.63** | 61.62 | 58.02 | 71.34 | 55.98 | 58.11 | 54.18 | 58.49 | 54.16 |
| DeiT-pre | 57.45 | 54.72 | 55.56 | 39.37 | 38.49 | 39.11 | 43.56 | 44.18 | 46.69 | 57.76 | 38.68 | 52.61 | **37.76** |
| EVA | 43.75 | 40.37 | 43.40 | **27.45** | 27.94 | 27.68 | 37.33 | 34.19 | 36.10 | 39.34 | 31.11 | 36.05 | 33.37 |
| Res-50 | 66.95 | 64.29 | 58.41 | 83.29 | 59.51 | 45.53 | 55.22 | 53.97 | 51.19 | 63.03 | 49.67 | 56.62 | **43.63** |
| Res-50 (SCL) | 50.06 | 97.59 | 98.46 | 77.52 | 48.18 | 31.57 | 98.80 | 38.47 | 32.78 | 96.02 | 45.06 | 97.58 | **25.04** |
| Average | 54.78 | 57.52 | 62.18 | 49.50 | 42.98 | 41.52 | 56.75 | 52.43 | 43.90 | 55.34 | 42.64 | 51.67 | **38.34** |

## 4.1 OOD Detection on ImageNet Benchmark

Table 1 presents an extensive evaluation of OOD detection performance on the ImageNet benchmark, spanning 9 diverse model backbones and 12 scoring methods. In contrast to prior work that typically centers evaluation on ResNet-50 Liu and Qin [2024], Sun et al. [2022], Park et al. [2023], Ren et al. [2021], Sun et al. [2021], Xu et al. [2024b], we expand the setting to include modern transformer-based architectures—ConvNeXt, Swin, DeiT, and EVA—as well as their ImageNet-21k pretrained variants when available. This large-scale comparison provides a better performance evaluation across architectures.

ORA consistently demonstrates strong performance across the board. It achieves the best average FPR95 across all models at 38.34%, outperforming the second-best method by a margin of 3.18% and the best FPR95 in 5 out of 9 model settings. Notably, ORA achieves the single best FPR95 score overall with 25.04% on the ResNet-50 (SCL) model, outperforming the next-best score of 27.45% obtained by R-Mah with EVA by a margin of 2.41%.

Furthermore, in the few cases where ORA is not the top-performing method—such as with vanilla Swin or DeiT models—there is no alternative method that clearly dominates. The best-performing method varies across these settings (R-Mah, R-Cos and P-Cos), indicating that no other baseline consistently adapts well to different backbones. In contrast, ORA remains within the top three methods across all model architectures, demonstrating both robustness and versatility. These results highlight ORA's ability to generalize across both convolutional and transformer-based models, pretrained or otherwise, and establish it as a strong state-of-the-art approach for OOD detection on ImageNet.

**Near-OOD Results.** In Appendix I we present complementary near-OOD evaluations on the NINCO [Bitterwolf et al., 2023] and SSB-Hard [Vaze et al., 2022] datasets. Because these benchmarks feature higher class similarity between ID and OOD samples, all methods show degraded separability compared to far-OOD settings and performance differences are more subtle. In this regime, approaches that explicitly use class centroids tend to be more effective (albeit with higher memory and information requirements), especially when OOD examples are visually close to certain ID classes. Although ORA does not rely on individual class means, it achieves competitive performance on par with centroid-based methods. Consistent with Bitterwolf et al. [2023], we also observe that no single method emerges as clearly state-of-the-art in this benchmark.

## 4.2 OOD Detection with CLIP

In this section, we evaluate the performance of post-hoc OOD detection methods on the CLIP architecture. CLIP, with its pre-trained vision-language capabilities, provides a foundation for OOD detection. We explore two approaches: linear probing and a zero-shot extension that leverages CLIP's inherent structure to avoid additional training.

**Linear probing with CLIP.** For methods requiring decision boundaries (e.g. Hendrycks and Gimpel [2022], Liu et al. [2020], Liu and Qin [2024], and ORA), we used the vision encoder of CLIP to extract features and trained a linear probe for the classification task. While effective, this reduces flexibility due to the need for additional training.

Table 2: OOD detection performance on the ImageNet benchmark using CLIP-ViT-H/14 features. We report FPR95 (%) and AUROC (%) for four OOD datasets under two regimes: linear probing (lp) and zero-shot (zs). The best result per column is highlighted in **bold**.

| Method | iNaturalist | | SUN | | Places | | Texture | | Avg | |
|--------|--------|--------|--------|--------|--------|--------|--------|--------|--------|--------|
| | FPR95↓ | AUROC↑ | FPR95↓ | AUROC↑ | FPR95↓ | AUROC↑ | FPR95↓ | AUROC↑ | FPR95↓ | AUROC↑ |
| MSP/lp | 15.74 | 96.64 | 46.00 | 88.68 | 48.73 | 87.40 | 40.87 | 87.98 | 37.83 | 90.18 |
| Energy/lp | 7.26 | 97.94 | 34.62 | 92.13 | 41.32 | 90.05 | 37.02 | 90.98 | 30.06 | 92.77 |
| MaxL/lp | 9.02 | 97.91 | 37.15 | 91.70 | 42.25 | 89.86 | 36.37 | 90.80 | 31.20 | 92.57 |
| NNG/lp | 7.26 | 97.94 | 34.62 | 92.12 | 41.32 | 90.05 | 37.02 | 90.98 | 30.06 | 92.77 |
| KNN/zs | 80.20 | 87.86 | 86.68 | 84.63 | 73.51 | 86.07 | 70.27 | 84.60 | 77.66 | 85.79 |
| fDBD/zs | 9.31 | 98.11 | 22.32 | 94.78 | 29.15 | 93.20 | 40.12 | 90.25 | 25.23 | 94.08 |
| fDBD/lp | 5.62 | 98.48 | 32.18 | 93.89 | 35.74 | 92.54 | 27.13 | 93.71 | 25.17 | 94.66 |
| Pcos/zs | 19.86 | 96.09 | 28.67 | 93.58 | 36.09 | 91.84 | 45.21 | 88.97 | 32.46 | 92.62 |
| MCM/zs | 17.81 | 96.52 | 26.04 | 93.94 | 35.07 | 91.54 | 41.12 | 90.92 | 30.01 | 93.23 |
| RMh/zs | **3.64** | **98.85** | 30.71 | 93.12 | 34.72 | 92.18 | **21.21** | **93.84** | 22.57 | 94.49 |
| ORA/zs | 14.12 | 97.41 | **22.97** | **94.97** | **28.01** | **93.41** | 38.28 | 90.73 | 25.85 | 94.13 |
| ORA/lp | 6.66 | 98.16 | 30.35 | 94.43 | 33.79 | 93.20 | 24.95 | 94.34 | 23.94 | **95.03** |

**Zero-shot OOD detection with CLIP.** To mitigate the limitation of requiring decision boundaries, we derive a zero-shot extension that defines them using text embeddings. For vision-language models like CLIP, we define the decision boundaries by vectors whose cosine similarity to the text embeddings of two classes are equal. Since cosine similarity is scale-invariant, these vectors are unique up to its norm. To compute the projection of a feature representation onto the decision boundary, we use normalized representations. Let $\mathbf{z}_{c_1}$ and $\mathbf{z}_{c_2}$ be the text embeddings of classes $c_1$ and $c_2$, and $\mathbf{z}$ be the feature representation. The decision boundary between $c_1$ and $c_2$ satisfies: (i) $\langle \mathbf{z}, \mathbf{z}_{c_1} \rangle = \langle \mathbf{z}, \mathbf{z}_{c_2} \rangle$, (ii) $\langle \mathbf{z}, (\mathbf{z}_{c_1} - \mathbf{z}_{c_2}) \rangle = 0$. Define $\mathbf{u} = \frac{(\mathbf{z}_{c_1} - \mathbf{z}_{c_2})}{||(\mathbf{z}_{c_1} - \mathbf{z}_{c_2})||_2}$. Then, the projection onto the decision boundary is $\mathbf{z}_{db} = \mathbf{z} - \langle \mathbf{z}, \mathbf{u} \rangle \cdot \mathbf{u}$. This projection enables zero-shot computation of the ORA score. Table 2 summarizes the performance of these methods in both linear probe and zero-shot settings. ORA achieves a strong performance on both linear probing and zero-shot settings with 23.94% and 25.85% FPR95 respectively.

**Positive effect of contrastive features.** An important observation from our results is the positive effect of contrastive learning on ORA's performance. Contrastive objectives—whether within a single modality, such as SCL, or across modalities, as in CLIP's image-text training—impose a geometric structure on the representation space that separates semantic concepts. ORA benefits particularly from this structure as confirmed by the results in Table 3: we show that ORA delivers the lowest average FPR95 on both benchmarks, cutting the previous best score from 11.85% to 10.97% on CIFAR-10 (-0.88

Table 3: Average OOD performance across CIFAR-10 (ResNet-18) and ImageNet (ResNet-50) benchmarks. All methods are evaluated using checkpoints trained with SCL. Reported with FPR95↓ and AUROC↑. Best results per column are in **bold**.

| Method | CIFAR-10 | | ImageNet | |
|--------|--------|--------|--------|--------|
| | FPR95↓ | AUROC↑ | FPR95↓ | AUROC↑ |
| SSD+ | 18.51 | 97.02 | 63.24 | 80.09 |
| KNN+ | 13.35 | 97.56 | 38.47 | 90.91 |
| fDBD | 11.85 | 97.60 | 32.78 | 92.86 |
| **ORA** | **10.97** | **97.67** | **25.04** | **94.26** |

pp) and from 32.78% to 25.04% on ImageNet (-7.74 pp), thereby outperforming every competing method with SCL-trained ResNet-18 and ResNet-50 backbones. Similarly, in Table 2, ORA shows strong results in both the zero-shot and linear probe settings of CLIP. These findings suggest that ORA is particularly well-suited to contrastively structured representations, hinting at a deeper connection between contrastive learning and OOD detection performance.

## 4.3 Model Ensembling with ORA

Recent works Xue et al. [2024] and Xu et al. [2024a] show that creating an ensemble of models can enhance the OOD performance. Inspired from these works, and from the observation that scale invariant representations are compatible between distinct models [Moschella et al., 2023], we demonstrate that *scale-invariant score functions can aggregate the confidences from different models*, by simply summing their scores. On Table 4 we show the individual performances of models ResNet-50, ResNet-50 with SCL and ViT-B/16 as well as their combined performances using the scale-invariant ORA.

Note that we demonstrate the scale-invariance property of fDBD in Equation 4 and include it in ensemble experiments to compare with ORA. It can be seen that for both of the score functions, the performance of ensemble is better than their individual counterparts showing that score aggregation improves their OOD performance. Moreover, the ensemble with ORA achieves a performance with $22.53\%$ FPR95 and $96.41\%$ AUROC, improving the metrics compared to the best individual performer in the ensemble by $2.51\%$ and $2.15\%$ respectively. In summary, we demonstrate that scale-invariance of ORA allows aggregating different models' confidences to solve OOD Detection Problem.

Table 4: ORA can be used for ensemble OOD detection due to its scale-invariance property. Evaluated on the ImageNet OOD benchmark. Best performance is highlighted in **bold**.

| Method | FPR95↓ | AUROC↑ |
|---|---|---|
| fDBD w/ResNet50 | 51.35 | 89.20 |
| fDBD w/ResNet50-supcon | 32.78 | 92.86 |
| fDBD w/ViT-B/16 | 41.55 | 91.05 |
| ORA w/ResNet50 | 44.58 | 90.68 |
| ORA w/ResNet50-supcon | 25.04 | 94.26 |
| ORA w/ViT-B/16 | 39.92 | 91.38 |
| Ensemble fDBD | 31.05 | 95.29 |
| Ensemble ORA | **22.53** | **96.41** |

## 4.4 ORA with Activation Shaping Algorithms

Recent methods ReAct Sun et al. [2021], ASH Djurisic et al. [2023] and Scale Xu et al. [2024b] show their success to modify the feature representations to reduce model's overconfident predictions. All three methods adopt a hyperparameter percentile to choose how to truncate and scale the feature representations using ID data statistics. When combined with Energy Liu et al. [2020] score, these methods improve the OOD Detection performance. On Table 5 we show that applying ORA scoring after activation shaping algorithms improves the performance. Specifically combining ORA with ReAct

Table 5: ORA can be used as a plug-in on top of activation shaping. Evaluated on the ImageNet OOD benchmark.

| Method | FPR95↓ | AUROC↑ |
|---|---|---|
| ORA w/ReLU | 25.04 | 94.26 |
| ORA w/ASH | 23.47 | 94.58 |
| ORA w/Scale | 23.34 | 94.37 |
| ORA w/ReAct | **20.36** | **96.29** |

reduces FPR95 from $25.04\%$ to $20.36\%$ highlighting both the flexibility and efficacy of our method. This demonstrates that ORA can flexibly be combined with activation shaping algorithms.

## 4.5 Ablation Studies

In this section, we will demonstrate the effectiveness of design choices on our score function ORA. We first justify our choice of centering in $\boldsymbol{\mu}_{\text{ID}}$ empirically, among the candidates: $\boldsymbol{\mu}_{\text{ID}}$, $\boldsymbol{\mu}_{y_{\text{pred}}}$, $\boldsymbol{\mu}_{y_{\text{target}}}$ and $\max(\mathbf{z}_{\text{ID}})$. Then, we compare different angle aggregation techniques across classes by replacing our $\max\left(\{\theta y, y'\}_{y' \in \mathcal{Y}, y' \neq y}\right)$ with mean and min across classes.

Table 6: Ablation on different centering strategies. Evaluated on both CIFAR-10 and ImageNet OOD benchmarks.

| Method | CIFAR-10 | | ImageNet | |
|---|---|---|---|---|
| | FPR95↓ | AUROC↑ | FPR95↓ | AUROC↑ |
| ORA w/ $\mu_{y_{\text{pred}}}$ | 12.42 | 97.59 | 43.02 | 89.86 |
| ORA w/ $\mu_{y_{\text{target}}}$ | 13.26 | 97.48 | 28.29 | 93.46 |
| ORA w/ $\max(\mathbf{z}_{\text{ID}})$ | 13.39 | 97.42 | 32.44 | 92.01 |
| ORA w/ $\mu_{\text{ID}}$ | **10.97** | **97.67** | **25.04** | **94.26** |

**Centering with $\mu_{\text{ID}}$ incorporates ID-statistics without biasing towards one particular class.** Table 6 shows the performance comparison between centerings with respect to different points. Using the relative angle with respect to the predicted ($\boldsymbol{\mu}_{y_{\text{pred}}}$) or target ($\boldsymbol{\mu}_{y_{\text{target}}}$) class centroid induce a bias towards the corresponding class, which in the end hinders the compatibility between angles coming across classes. On the other hand, using $\max(\mathbf{z}_{\text{ID}})$ shifts every feature representation to the same orthant, reducing to simply computing the absolute distance between feature representations and the decision boundaries, which is agnostic from the in-distribution feature statistics. We observe a

Table 7: Ablation on the different score aggregations across classes. Evaluated under both ImageNet and CIFAR-10 OOD benchmarks.

| Method | CIFAR-10 | | ImageNet | |
|---|---|---|---|---|
| | FPR95↓ | AUROC↑ | FPR95↓ | AUROC↑ |
| ORA w/ min | 32.02 | 95.23 | 79.15 | 81.38 |
| ORA w/ mean | 11.84 | 97.59 | 32.76 | 92.87 |
| ORA w/ max | **10.97** | **97.67** | **25.04** | **94.26** |

significant improvement in performance when computing relative angles using $\mu_{\mathrm{ID}}$, demonstrating the importance of incorporating in-distribution (ID) statistics when measuring the relationship between feature representations and decision boundaries. ORA with $\mu_{\mathrm{ID}}$ centering improves the FPR95 by up to $1.45\%$ and $7.4\%$ on CIFAR-10 and ImageNet respectively while also improving the AUROC for both benchmarks.

**Looking at the furthest class is better for ID/OOD separation.** On Table 7 we explored different ways to aggregate class specific angles. Originally, we devise our score function to return the maximum relative angle discrepancy between the feature representation across decision boundaries. Intuitively, this suggests that considering the furthest possible class that a feature belongs from the mean of in-distribution's perspective is effective to distinguish OOD from ID. On the other hand, comparing the minimum focuses on the smallest relative angle, reducing the separability significantly. Table 7 demonstrates taking the maximum across classes clearly outperforms mean and min aggregations, improving FPR95 and AUROC metrics on both benchmarks. Specifically the difference is higher on our large-scale experiments reducing the FPR95 by $7.72\%$ and increasing the AUROC by $1.39\%$ compared to the second best aggregation.

## 5   Conclusion

In this paper, we introduce a novel angle-based OOD detection score function. As a post-hoc measure of model confidence, ORA offers several key advantages: it is (i) hyperparameter-free, (ii) model-agnostic and (iii) scale-invariant. These features allow ORA to be applied to arbitrary pretrained models and used in conjunction with existing activation shaping algorithms, enhancing the performance. Notably, its scale-invariant nature enables simple aggregation of multiple models' confidences through score summation, allowing a creation of an effective model ensemble for OOD detection. Our extensive experiments demonstrate that ORA achieves state-of-the-art performance, using the relationship between the feature representations and decision boundaries relative to the ID statistics effectively.

Beyond the global ID mean, we tested class-centered variants (per-class means and predicted-class mean) and found no consistent gains. A promising future direction is a data-driven weighted combination of per-class angle scores, with weights derived from ID statistics (e.g. inverse class-wise feature variance or proportional to class accuracy), preserving ORA's hyperparameter-free character while emphasizing reliable anchors. We leave a systematic study of this extension to future work.

## Acknowledgements

Francesco Locatello's contribution to this research was funded in part by the Austrian Science Fund (FWF) 10.55776/COE12.

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

Table 8: ORA achieves state-of-the-art performance on CIFAR-10 OOD benchmark. Evaluated on ResNet-18 with FPR95 and AUROC. ↑ indicates that larger values are better and vice versa. Best performance highlighted in **bold**. Methods with * are hyperparameter-free.

| Method | SVHN | | iSUN | | Place365 | | Texture | | Avg | |
|---|---|---|---|---|---|---|---|---|---|---|
| | FPR95↓ | AUROC↑ | FPR95↓ | AUROC↑ | FPR95↓ | AUROC↑ | FPR95↓ | AUROC↑ | FPR95↓ | AUROC↑ |
| *Without Contrastive Learning* | | | | | | | | | | |
| MSP * | 59.51 | 91.29 | 54.57 | 92.12 | 62.55 | 88.63 | 66.49 | 88.50 | 60.78 | 90.14 |
| ODIN | 61.71 | 89.12 | 15.09 | 97.37 | 41.45 | 91.85 | 52.62 | 89.41 | 42.72 | 91.94 |
| Energy * | 53.96 | 91.32 | 27.52 | 95.59 | 42.80 | 91.03 | 55.23 | 89.37 | 44.88 | 91.83 |
| ViM | 25.38 | 95.40 | 30.52 | 95.10 | 47.36 | 90.68 | 25.69 | 95.01 | 32.24 | 94.05 |
| MDS | 16.77 | 95.67 | 7.56 | 97.93 | 85.87 | 68.44 | 35.21 | 85.90 | 36.35 | 86.99 |
| *With Contrastive Learning* | | | | | | | | | | |
| CSI | 37.38 | 94.69 | 10.36 | 98.01 | 38.31 | 93.04 | 28.85 | 94.87 | 28.73 | 95.15 |
| SSD+ | 1.35 | 99.72 | 33.60 | 95.16 | 26.09 | 95.48 | 12.98 | 97.70 | 18.51 | 97.02 |
| KNN+ | 2.20 | 99.57 | 20.06 | 96.74 | 23.06 | 95.36 | 8.09 | 98.56 | 13.35 | 97.56 |
| fDBD * | 4.59 | 99.00 | 10.04 | 98.07 | 23.16 | 95.09 | 9.61 | 98.22 | 11.85 | 97.60 |
| ORA * | 3.53 | 99.16 | 8.36 | 98.28 | 23.40 | 94.88 | 8.58 | 98.34 | **10.97** | **97.67** |

Table 9: ORA achieves state-of-the-art performance on ImageNet OOD benchmark. Evaluated on ResNet-50 with FPR95 and AUROC. ↑ indicates that larger values are better and vice versa. Best performance highlighted in **bold**. Methods with * are hyperparameter-free.

| Method | iNaturalist | | SUN | | Places | | Texture | | Avg | |
|---|---|---|---|---|---|---|---|---|---|---|
| | FPR95↓ | AUROC↑ | FPR95↓ | AUROC↑ | FPR95↓ | AUROC↑ | FPR95↓ | AUROC↑ | FPR95↓ | AUROC↑ |
| *Without Contrastive Learning* | | | | | | | | | | |
| MSP * | 54.99 | 87.74 | 70.83 | 80.63 | 73.99 | 79.76 | 68.00 | 79.61 | 66.95 | 81.99 |
| ODIN | 47.66 | 89.66 | 60.15 | 84.59 | 67.90 | 81.78 | 50.23 | 85.62 | 56.48 | 85.41 |
| Energy * | 55.72 | 89.95 | 59.26 | 85.89 | 64.92 | 82.86 | 53.72 | 85.99 | 58.41 | 86.17 |
| ViM | 71.85 | 87.42 | 81.79 | 81.07 | 83.12 | 78.40 | 14.88 | 96.83 | 62.91 | 85.93 |
| MDS | 97.00 | 52.65 | 98.50 | 42.41 | 98.40 | 41.79 | 55.80 | 85.01 | 87.43 | 55.17 |
| *With Contrastive Learning* | | | | | | | | | | |
| SSD+ | 57.16 | 87.77 | 78.23 | 73.10 | 81.19 | 70.97 | 36.37 | 88.53 | 63.24 | 80.09 |
| KNN+ | 30.18 | 94.89 | 48.99 | 88.63 | 59.15 | 84.71 | 15.55 | 95.40 | 38.47 | 90.91 |
| fDBD * | 17.27 | 96.68 | 42.30 | 90.90 | 49.77 | 88.36 | 21.83 | 95.43 | 32.78 | 92.86 |
| ORA * | 12.27 | 97.42 | 31.80 | 92.85 | 40.71 | 90.10 | 15.39 | 96.68 | **25.04** | **94.26** |

# A  Additional Results

## A.1  OOD Detection on CIFAR-10 and ImageNet Benchmarks

Table 8 and Table 9 shows the performance of ORA along with the 9 baselines on CIFAR-10 and ImageNet OOD Benchmarks, respectively. All the baselines on CIFAR-10 use ResNet-18 architecture and on ImageNet use ResNet-50. Our proposed method reaches *state-of-the-art* performance on both benchmarks, reducing the FPR95 on average by 7.74% on Imagenet and 0.88% on CIFAR10. In the following, we provide a detailed analysis of these results.

**ORA continues to show the success of distance-based methods over logit-based methods.** Logit-based scoring methods MSP Hendrycks and Gimpel [2022], Energy Liu et al. [2020] are one of the earliest baselines proving their success on measuring model's confidences. MSP measures the maximum softmax probability as its score while Energy does a logsumexp operation on the logits. Recent distance-based methods like KNN+ Sun et al. [2022] and fDBD Liu and Qin [2024] outperforms the early logit-based ones. Similarly, ORA achieves significantly better performance on both benchmarks, reducing the FPR95 up to 49.81% and 41.91% while improving the AUROC up to 7.53% and 12.27% on CIFAR-10 and ImageNet OOD benchmarks.

**ORA improves on the recent success of methods using contrastively learned features.** Table 8 and 9 show the success of recent methods CSI Tack et al. [2020], SSD+ Sehwag et al. [2021], KNN+ Sun et al. [2022] and fDBD Liu and Qin [2024] that utilizes contrastively learned representations over the ones those do not use. We observe that the additional structure the supervised contrastive loss puts on the feature representations are particularly beneficial to the distance-based methods. ORA also benefits from more structured representations on the feature space, as it explores the relationship between the representation and the decision boundaries. Notably, ORA improves both of the metrics on both CIFAR-10 and ImageNet benchmarks, achieving the state-of-the-art performance.

## A.2 Comparison with ReAct

Tables 10 and 11 compare the performance of ORA with the activation shaping method ReAct Sun et al. [2021]. ORA improves FPR95 by $1.11\%$ on CIFAR10 and $6.39\%$ on ImageNet. Additionally, combining ORA with ReAct further enhances performance on both benchmarks across both metrics, FPR95 and AUROC.

Table 10: ORA vs ReAct under ImageNet OOD benchmark.

| Method | iNaturalist | | SUN | | Places | | Texture | | Avg | |
|---|---|---|---|---|---|---|---|---|---|---|
| | FPR95↓ | AUROC↑ | FPR95↓ | AUROC↑ | FPR95↓ | AUROC↑ | FPR95↓ | AUROC↑ | FPR95↓ | AUROC↑ |
| ReAct | 20.38 | 96.22 | 24.20 | 94.20 | 33.85 | 91.58 | 47.30 | 89.80 | 31.43 | 92.95 |
| ORA | 12.27 | 97.42 | 31.80 | 92.85 | 40.71 | 90.10 | 15.39 | 96.68 | 25.04 | 94.26 |
| ORA w/ReAct | 11.13 | 97.79 | 22.34 | 94.95 | 33.33 | 91.81 | 14.65 | 96.60 | **20.36** | **96.29** |

Table 11: ORA vs ReAct under CIFAR OOD benchmark.

| Method | SVHN | | iSUN | | Places | | Texture | | Avg | |
|---|---|---|---|---|---|---|---|---|---|---|
| | FPR95↓ | AUROC↑ | FPR95↓ | AUROC↑ | FPR95↓ | AUROC↑ | FPR95↓ | AUROC↑ | FPR95↓ | AUROC↑ |
| ReAct | 6.15 | 98.75 | 10.31 | 98.09 | 21.68 | 95.47 | 10.18 | 98.12 | 12.08 | 97.61 |
| ORA | 3.53 | 99.16 | 8.36 | 98.28 | 23.40 | 94.88 | 8.58 | 98.34 | 10.97 | 97.67 |
| ORA w/ReAct | 3.35 | 99.18 | 8.11 | 98.29 | 20.84 | 95.25 | 7.87 | 98.45 | **10.04** | **97.79** |

## A.3 Centering with Different Statistics

Tables 12 and 13 present the performance of the ORA score when using different class means as the reference view instead of the mean of ID features. We also explored alternative centering strategies by replacing the mean of ID features with elementwise operations—max, min, and median—where each corresponds to using the respective statistic before calculating angles. Additionally, sum aggregation refers to summing scores obtained from individual class mean reference points or from these elementwise operations. Results indicate that using $\mu_{\text{ID}}$ consistently outperforms these alternatives.

Table 12: CIFAR10 centering with different statistics using ResNet18 model.

| Method | SVHN | | iSUN | | Places | | Texture | | Avg | | ID Acc |
|---|---|---|---|---|---|---|---|---|---|---|---|
| | FPR95↓ | AUROC↑ | FPR95↓ | AUROC↑ | FPR95↓ | AUROC↑ | FPR95↓ | AUROC↑ | FPR95↓ | AUROC↑ | |
| Class 0 mean | 4.77 | 98.96 | 8.06 | 98.27 | 25.20 | 94.62 | 10.11 | 98.19 | 12.03 | 97.51 | 95.2 |
| Class 1 mean | 6.12 | 98.77 | 8.86 | 98.24 | 24.94 | 94.96 | 13.16 | 97.68 | 13.27 | 97.41 | 98.5 |
| Class 2 mean | 5.42 | 98.84 | 7.90 | 98.36 | 22.19 | 95.58 | 11.42 | 97.98 | 11.73 | 97.69 | 92.6 |
| Class 3 mean | 5.94 | 98.76 | 7.99 | 98.29 | 22.80 | 95.43 | 11.35 | 97.66 | 12.02 | 97.54 | 88.7 |
| Class 4 mean | 5.44 | 98.85 | 8.87 | 98.22 | 22.68 | 95.47 | 11.26 | 97.96 | 12.06 | 97.63 | 95.8 |
| Class 5 mean | 6.22 | 98.64 | 7.38 | 98.45 | 23.11 | 95.48 | 11.84 | 97.82 | 12.14 | 97.60 | 88.0 |
| Class 6 mean | 5.74 | 98.82 | 8.50 | 98.26 | 97.67 | 20.76 | 12.11 | 95.73 | 11.78 | 97.62 | 97.9 |
| Class 7 mean | 5.93 | 98.78 | 8.29 | 98.30 | 24.55 | 95.13 | 12.57 | 97.82 | 12.84 | 97.51 | 96.4 |
| Class 8 mean | 5.81 | 98.81 | 10.03 | 97.95 | 26.79 | 94.18 | 10.41 | 98.11 | 13.26 | 97.26 | 97.0 |
| Class 9 mean | 6.11 | 98.78 | 9.00 | 98.19 | 24.89 | 94.62 | 11.35 | 97.95 | 12.84 | 97.25 | 96.3 |
| Sum aggregation | 5.68 | 98.85 | 8.27 | 98.33 | 23.70 | 95.34 | 11.33 | 97.98 | 12.25 | 97.62 | - |
| Elementwise max | 6.28 | 98.73 | 8.28 | 98.30 | 13.79 | 97.57 | 24.35 | 95.21 | 13.18 | 97.45 | - |
| Elementwise min | 3.60 | 99.14 | 14.82 | 97.10 | 9.38 | 97.99 | 27.62 | 92.97 | 13.85 | 96.80 | - |
| Elementwise median | 2.33 | 99.34 | 10.02 | 97.90 | 7.73 | 98.29 | 23.99 | 93.84 | 11.02 | 97.34 | - |
| Sum aggregation | 5.78 | 98.65 | 20.31 | 95.64 | 10.35 | 97.80 | 30.42 | 91.60 | 16.72 | 95.92 | - |
| ORA | 3.53 | 99.16 | 8.36 | 98.28 | 23.40 | 94.88 | 8.58 | 98.34 | **10.97** | **97.67** | 94.6 |

Table 13: ImageNet centering with different statistics using ResNet50 model.

| Method | iNaturalist | | SUN | | Places | | Texture | | Avg | | ID Acc |
|---|---|---|---|---|---|---|---|---|---|---|---|
| | FPR95↓ | AUROC↑ | FPR95↓ | AUROC↑ | FPR95↓ | AUROC↑ | FPR95↓ | AUROC↑ | FPR95↓ | AUROC↑ | |
| Class 1 mean | 16.01 | 96.92 | 31.63 | 92.52 | 39.86 | 90.67 | 25.39 | 93.33 | 28.22 | 93.36 | 92 |
| Class 250 mean | 11.48 | 97.65 | 31.20 | 92.69 | 39.53 | 90.77 | 20.16 | 94.87 | 25.59 | 93.99 | 66 |
| Class 500 mean | 14.87 | 97.08 | 38.97 | 90.52 | 45.80 | 88.90 | 26.29 | 93.04 | 31.48 | 92.28 | 60 |
| Class 750 mean | 11.57 | 97.59 | 34.23 | 92.13 | 42.60 | 90.15 | 19.75 | 95.18 | 27.04 | 93.76 | 84 |
| Class 1000 mean | 11.36 | 97.63 | 30.20 | 93.01 | 38.12 | 91.08 | 19.31 | 95.31 | **24.75** | **94.26** | 60 |
| Sum aggregation | 12.40 | 97.48 | 32.47 | 92.36 | 40.42 | 90.53 | 21.38 | 94.52 | 26.67 | 93.72 | - |
| Elementwise Max | 17.10 | 96.76 | 34.22 | 91.73 | 41.88 | 90.14 | 35.09 | 89.84 | 32.07 | 92.12 | - |
| Elementwise Min | 29.16 | 94.51 | 60.70 | 85.81 | 65.01 | 83.18 | 22.84 | 95.07 | 44.43 | 89.64 | - |
| Elementwise Median | 20.04 | 95.83 | 46.66 | 89.15 | 54.42 | 85.61 | 15.04 | 96.81 | 34.04 | 91.85 | - |
| Sum aggregation | 21.09 | 95.89 | 49.47 | 88.97 | 56.50 | 86.12 | 17.62 | 95.98 | 36.17 | 91.74 | - |
| ORA | 12.27 | 97.42 | 31.80 | 92.85 | 40.71 | 90.10 | 15.39 | 96.68 | 25.04 | **94.26** | 77.33 |

## A.4 Resource Constrained Setting

Table 14 presents the performance of ORA in a resource-constrained setting using MobileNetV2 Sandler et al. [2018], a model designed for efficient inference with minimal resources. ORA achieves the best average performance, improving FPR and AUROC by 7.56% and 1.66%, respectively, demonstrating its adaptability to resource-limited scenarios.

Table 14: Resource Constrained Setting: ImageNet MobileNet_v2 performances.

| Method | iNaturalist | | SUN | | Places | | Texture | | Avg | |
|---|---|---|---|---|---|---|---|---|---|---|
| | FPR95↓ | AUROC↑ | FPR95↓ | AUROC↑ | FPR95↓ | AUROC↑ | FPR95↓ | AUROC↑ | FPR95↓ | AUROC↑ |
| MSP | 59.84 | 86.71 | 74.15 | 78.87 | 76.84 | 78.14 | 70.98 | 78.95 | 70.45 | 80.67 |
| Energy | 55.35 | 90.33 | 59.36 | 86.24 | 66.28 | 83.21 | 54.54 | 86.58 | 58.88 | 86.59 |
| KNN | 85.92 | 72.67 | 90.51 | 65.39 | 93.21 | 60.08 | 14.04 | 96.98 | 70.92 | 73.78 |
| fDBD | 53.72 | 90.89 | 68.22 | 82.84 | 73.20 | 80.09 | 37.82 | 91.85 | 58.24 | 86.42 |
| ORA | 46.59 | 91.86 | 61.21 | 85.01 | 67.81 | 82.08 | 27.07 | 94.04 | **50.68** | **88.25** |

## B Implementation Details

We used Pytorch [Paszke et al., 2019] to conduct our experiments. We obtain the checkpoints of pretrained models ResNet18 with supervised contrastive loss and ResNet50 with supervised contrastive loss from Liu and Qin [2024]'s work for a fair comparison. In the experiment where we aggregate different models' confidences, ViT-B/16 [Dosovitskiy et al., 2020] checkpoint is retrieved from the publicly available repository https://github.com/lukemelas/PyTorch-Pretrained-ViT/tree/master. In the experiment where we merge ORA with the activation shaping algorithms ASH [Djurisic et al., 2023], Scale [Xu et al., 2024b] and ReAct [Sun et al., 2021], we used the percentiles to set the thresholds 35, 90 and 80 respectively. For the extended results on Table 1, we used the timm [Wightman, 2020] checkpoints for the models ConvNeXt [Liu et al., 2022], Swin [Liu et al., 2021], DeiT [Touvron et al., 2021] and EVA [Fang et al., 2023]. Similarly, for the CLIP experiments on Table 2, we used the huggingface checkpoint of CLIP ViT-H/14 [LAION]. All experiments are evaluated on a single Nvidia H100 GPU. Note that, thanks to our hyperparameter-free post-hoc score function, all experiments are deterministic given the pretrained model.

## C Robustness of the Relative Angles

In this section, we provide a theoretical justification for focusing on relative angles, rather than absolute distances, to better understand the robustness of representations under scaling transformations.

**Theorem:** Let $M_1$ and $M_2$ be two neural networks such that:

1. The encoder of $M_2$ is a scaled version of the encoder of $M_1$. Specifically, there exists a positive scalar $k \in \mathbb{R}_+$ such that the output of the penultimate layer of $M_2$ satisfies:

$$\mathbf{z}^{M_2} = k \cdot \mathbf{z}^{M_1}$$

2. Both networks share the same final linear layer (without bias) for classification.

Since scaling transformations do not affect the softmax decision boundaries (due to monotonicity), $M_1$ and $M_2$ will produce the same classification decisions. Under this setup:

- The angles satisfy $\theta_{y_1,y_2}(\mathbf{z}^{M_1}) = \theta_{y_1,y_2}(\mathbf{z}^{M_2})$, where $\theta_{y_1,y_2}(\mathbf{z})$ is the relative angle between representation and its projection onto the decision boundary.

- However, the distances satisfy $d_{y_1,y_2}(\mathbf{z}^{M_2}, \mathbf{z}^{M_2}_{db}) = k \cdot d_{y_1,y_2}(\mathbf{z}^{M_1}, \mathbf{z}^{M_1}_{db})$, where $d(\cdot, \cdot)$ is the Euclidean distance.

**Proof:** Let $\mathbf{z}^{M_1}$ be the penultimate layer representation of $M_1$ and let $\mathbf{z}^{M_1}_{db}$ denote the projection of $\mathbf{z}^{M_1}$ onto the decision boundary between two classes $y_1$ and $y_2$. Then following the Equation 1

$$\mathbf{z}^{M_1}_{db} = \mathbf{z}^{M_1} - \frac{(\mathbf{w}_{y_1} - \mathbf{w}_{y_2})^T \mathbf{z}^{M_1}}{\|\mathbf{w}_{y_1} - \mathbf{w}_{y_2}\|^2}(\mathbf{w}_{y_1} - \mathbf{w}_{y_2})$$

where $\mathbf{w}_{y_1}$ and $\mathbf{w}_{y_2}$ are the weight vectors corresponding to classes $y_1$ and $y_2$ in the shared linear layer. For $M_2$, the features are scaled by $k$, so $\mathbf{z}^{M_2} = k \cdot \mathbf{z}^{M_1}$. Substituting into the projection formula:

$$\mathbf{z}^{M_2}_{db} = k \cdot \mathbf{z}^{M_1} - \frac{(\mathbf{w}_{y_1} - \mathbf{w}_{y_2})^T k \cdot \mathbf{z}^{M_1}}{\|\mathbf{w}_{y_1} - \mathbf{w}_{y_2}\|^2}(\mathbf{w}_{y_1} - \mathbf{w}_{y_2})$$

$$\mathbf{z}^{M_2}_{db} = k \cdot \mathbf{z}^{M_1}_{db}$$

The angle is defined by the cosine similarity between the vectors $\mathbf{z}^{M_1} - \boldsymbol{\mu}^{M_1}_{\text{ID}}$ and $\mathbf{z}^{M_1}_{db} - \boldsymbol{\mu}^{M_1}_{\text{ID}}$. Using the substitutions $\mathbf{z}^{M_2} = k \cdot \mathbf{z}^{M_1}$, $\mathbf{z}^{M_2}_{db} = k \cdot \mathbf{z}^{M_1}_{db}$, and $\boldsymbol{\mu}^{M_2}_{\text{ID}} = k \cdot \boldsymbol{\mu}^{M_1}_{\text{ID}}$, we have:

$$
\begin{aligned}
\cos\left(\theta_{y_1,y_2}(\mathbf{z}^{M_1})\right) &= \frac{\langle \mathbf{z}^{M_1} - \boldsymbol{\mu}^{M_1}_{\text{ID}}, \mathbf{z}^{M_1}_{db} - \boldsymbol{\mu}^{M_1}_{\text{ID}} \rangle}{\|\mathbf{z}^{M_1} - \boldsymbol{\mu}^{M_1}_{\text{ID}}\| \cdot \|\mathbf{z}^{M_1}_{db} - \boldsymbol{\mu}^{M_1}_{\text{ID}}\|} \\
&= \frac{k \cdot k \cdot \langle \mathbf{z}^{M_1} - \boldsymbol{\mu}^{M_1}_{\text{ID}}, \mathbf{z}^{M_1}_{db} - \boldsymbol{\mu}^{M_1}_{\text{ID}} \rangle}{k \cdot k \cdot \|\mathbf{z}^{M_1} - \boldsymbol{\mu}^{M_1}_{\text{ID}}\| \cdot \|\mathbf{z}^{M_1}_{db} - \boldsymbol{\mu}^{M_1}_{\text{ID}}\|} \\
&= \frac{\langle k \cdot \mathbf{z}^{M_1} - k \cdot \boldsymbol{\mu}^{M_1}_{\text{ID}}, k \cdot \mathbf{z}^{M_1}_{db} - k \cdot \boldsymbol{\mu}^{M_1}_{\text{ID}} \rangle}{\|k \cdot \mathbf{z}^{M_1} - k \cdot \boldsymbol{\mu}^{M_1}_{\text{ID}}\| \cdot \|k \cdot \mathbf{z}^{M_1}_{db} - k \cdot \boldsymbol{\mu}^{M_1}_{\text{ID}}\|} \\
&= \frac{\langle \mathbf{z}^{M_2} - \boldsymbol{\mu}^{M_2}_{\text{ID}}, \mathbf{z}^{M_2}_{db} - \boldsymbol{\mu}^{M_2}_{\text{ID}} \rangle}{\|\mathbf{z}^{M_2} - \boldsymbol{\mu}^{M_2}_{\text{ID}}\| \cdot \|\mathbf{z}^{M_2}_{db} - \boldsymbol{\mu}^{M_2}_{\text{ID}}\|} \\
&= \cos\left(\theta_{y_1,y_2}(\mathbf{z}^{M_2})\right)
\end{aligned}
$$

On the other hand, the absolute distance between $z$ and $z_{db}$ scale as follows:

$$
\begin{aligned}
d_{y_1,y_2}(\mathbf{z}^{M_2}, \mathbf{z}^{M_2}_{db}) &= d_{y_1,y_2}(k \cdot \mathbf{z}^{M_1}, k \cdot \mathbf{z}^{M_1}_{db}) \\
&= k \cdot d_{y_1,y_2}(\mathbf{z}^{M_1}, \mathbf{z}^{M_1}_{db})
\end{aligned}
$$

Therefore, for two networks $M_1$ and $M_2$ with identical performance, we demonstrate that relative angles remain invariant to scaling, whereas absolute distances are sensitive to it. Given that ReLU networks are commonly used in practice (where activations are unbounded), scale-invariant, angle-based techniques provide a more robust and suitable approach for measuring confidence compared to distance-based methods, especially when comparing the confidences of different models.

**Algorithm 1** ORA (OOD Detection with Relative Angles)

---

**Require:** Sample $\mathbf{x}$, Pretrained model $f$, Mean of the in-distribution features $\boldsymbol{\mu}_{\text{ID}}$
**Ensure:** OOD score $s$
 1: **function** ORA($\mathbf{x}, f, \boldsymbol{\mu}_{\text{ID}}$)
 2:     $\hat{y} \leftarrow \arg\max_{y \in \mathcal{Y}} f(\mathbf{x})$
 3:     $\mathbf{z} = f_1 \circ \ldots \circ f_{L-1}(\mathbf{x})$                                       ▷ penultimate layer features
 4:     $score \leftarrow -\infty$
 5:     **for** $y' \in \mathcal{Y}$ **and** $y' \neq \hat{y}$ **do**                             ▷ for each other class
 6:         compute $\mathbf{z}_{db}$ as in Eq. 1
 7:         compute $\theta_{\hat{y}, y'}(\mathbf{z})$ using Eq. 2
 8:         compute $\tilde{s}(\mathbf{z})$ using Eq. 3
 9:         **if** $\tilde{s}(\mathbf{z}) \geq score$ **then**
10:             $score = \tilde{s}(\mathbf{z})$
11:         **end if**
12:     **end for**
13:     **return** $score$                                  ▷ maximum score across classes
14: **end function**

---

## D   Algorithm Box

We present the pseudocode for ORA in Algorithm Box 1. It depicts how ORA assigns a score given a sample $x$, pretrained model $f$, and the ID statistics, mean of the in-distribution features $\mu_{\text{ID}}$.

## E   Detailed Tables and Figures

Table 15: Extended version for the model ensemble experiment presented on Table 4. ORA can be used as a score function to accumulate different architectures' confidences due to its scale-invariance property. Evaluated under both ImageNet OOD benchmark. Best performance highlighted in **bold**.

| Method | iNaturalist | | SUN | | Places | | Texture | | Avg | |
|---|---|---|---|---|---|---|---|---|---|---|
| | FPR95↓ | AUROC↑ | FPR95↓ | AUROC↑ | FPR95↓ | AUROC↑ | FPR95↓ | AUROC↑ | FPR95↓ | AUROC↑ |
| fDBD w/ResNet50 | 40.10 | 93.70 | 60.89 | 86.86 | 66.75 | 84.14 | 37.66 | 92.09 | 51.35 | 89.20 |
| fDBD w/ResNet50-supcon | 17.34 | 96.68 | 42.26 | 90.92 | 49.68 | 88.38 | 21.84 | 95.44 | 32.78 | 92.86 |
| fDBD w/ViT-B/16 | 12.97 | 97.71 | 51.09 | 89.67 | 56.51 | 87.32 | 45.62 | 89.48 | 41.55 | 91.05 |
| ORA w/ResNet50 | 34.88 | 94.43 | 54.30 | 88.41 | 61.79 | 85.64 | 27.34 | 94.24 | 44.58 | 90.68 |
| ORA w/ResNet50-supcon | 12.27 | 97.42 | 31.80 | 92.85 | 40.71 | 90.10 | 15.39 | 96.68 | 25.04 | 94.26 |
| ORA w/ViT-B/16 | 11.81 | 97.85 | 48.98 | 90.06 | 54.60 | 87.75 | 44.31 | 89.85 | 39.92 | 91.38 |
| Ensemble fDBD | 4.58 | 98.93 | 42.81 | 93.97 | 53.49 | 91.92 | 23.33 | 96.34 | 31.05 | 95.29 |
| Ensemble ORA | 2.77 | 99.29 | 30.21 | 95.39 | 42.52 | 93.39 | 14.63 | 97.59 | **22.53** | **96.41** |

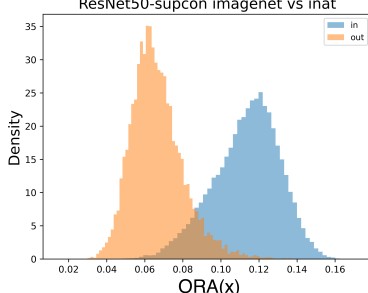 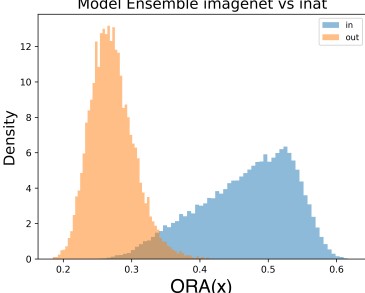

Figure 3: Comparison of the score histograms on Imagenet (ID) and inaturalist[Van Horn et al., 2018](OOD) of the best individual model (left) with the model ensemble (*right*). Model ensemble improves the ID and OOD separation.

Table 16: Extended version for the activation shaping experiment presented on Table 5. ORA can be used as a plug-in on top of activation shaping algorithms. Evaluated under ImageNet OOD benchmark. ↑ indicates that larger values are better and vice versa. Best performance highlighted in **bold**.

| Method | iNaturalist | | SUN | | Places | | Texture | | Avg | |
|---|---|---|---|---|---|---|---|---|---|---|
| | FPR95↓ | AUROC↑ | FPR95↓ | AUROC↑ | FPR95↓ | AUROC↑ | FPR95↓ | AUROC↑ | FPR95↓ | AUROC↑ |
| ORA w/ReLU | 12.27 | 97.42 | 31.80 | 92.85 | 40.71 | 90.10 | 15.39 | 96.68 | 25.04 | 94.26 |
| ORA w/ASH | 11.08 | 97.68 | 27.81 | 93.59 | 36.53 | 91.36 | 18.48 | 96.70 | 23.47 | 94.58 |
| ORA w/Scale | 14.65 | 97.05 | 25.43 | 94.02 | 36.21 | 90.78 | 17.07 | 95.65 | 23.34 | 94.37 |
| ORA w/ReAct | 11.13 | 97.79 | 22.34 | 94.95 | 33.33 | 91.81 | 14.65 | 96.60 | **20.36** | **96.29** |

# F Plain Model Performances

Tables 17, 18, and 19 show the performance of feature-based OOD methods on models trained without supervised contrastive loss. The results highlight that supervised contrastive loss significantly enhances feature quality, leading to a substantial performance boost for feature-based OOD methods.

Table 17: CIFAR10 Plain ResNet18 performances.

| Method | SVHN | | iSUN | | Places | | Texture | | Avg | |
|---|---|---|---|---|---|---|---|---|---|---|
| | FPR95↓ | AUROC↑ | FPR95↓ | AUROC↑ | FPR95↓ | AUROC↑ | FPR95↓ | AUROC↑ | FPR95↓ | AUROC↑ |
| KNN | 27.85 | 95.52 | 24.67 | 95.52 | 44.56 | 90.85 | 37.57 | 94.71 | 33.66 | 94.15 |
| fDBD | 22.58 | 96.07 | 23.96 | 95.85 | 46.59 | 90.40 | 31.24 | 94.48 | 31.09 | 94.20 |
| ORA | 22.09 | 96.02 | 22.91 | 95.90 | 46.46 | 90.37 | 31.28 | 94.48 | **30.86** | **94.21** |

Table 18: ImageNet Plain ResNet50 performances.

| Method | iNaturalist | | SUN | | Places | | Texture | | Avg | |
|---|---|---|---|---|---|---|---|---|---|---|
| | FPR95↓ | AUROC↑ | FPR95↓ | AUROC↑ | FPR95↓ | AUROC↑ | FPR95↓ | AUROC↑ | FPR95↓ | AUROC↑ |
| KNN | 59.00 | 86.47 | 68.82 | 80.72 | 76.28 | 75.76 | 11.77 | 97.07 | 53.97 | 85.01 |
| fDBD | 40.24 | 93.67 | 60.60 | 86.97 | 66.40 | 84.27 | 37.50 | 92.12 | 51.19 | **89.26** |
| ORA | 38.94 | 93.68 | 59.78 | 86.53 | 66.89 | 83.04 | 31.67 | 93.33 | **49.32** | 89.15 |

Table 19: ImageNet ViT performances.

| Method | iNaturalist | | SUN | | Places | | Texture | | Avg | |
|---|---|---|---|---|---|---|---|---|---|---|
| | FPR95↓ | AUROC↑ | FPR95↓ | AUROC↑ | FPR95↓ | AUROC↑ | FPR95↓ | AUROC↑ | FPR95↓ | AUROC↑ |
| KNN | 11.41 | 97.65 | 56.91 | 86.39 | 63.76 | 82.61 | 42.23 | 89.61 | 43.58 | 89.07 |
| fDBD | 12.86 | 97.72 | 50.86 | 89.74 | 56.28 | 87.44 | 45.74 | 89.41 | 41.44 | 91.08 |
| ORA | 11.80 | 97.86 | 48.81 | 90.14 | 54.32 | 87.88 | 44.56 | 89.75 | **39.87** | **91.41** |

# G  Histogram Plots for ID/OOD Separability

Figures 4 and 5 shows the score distributions on the Tables 8 and 9 respectively.

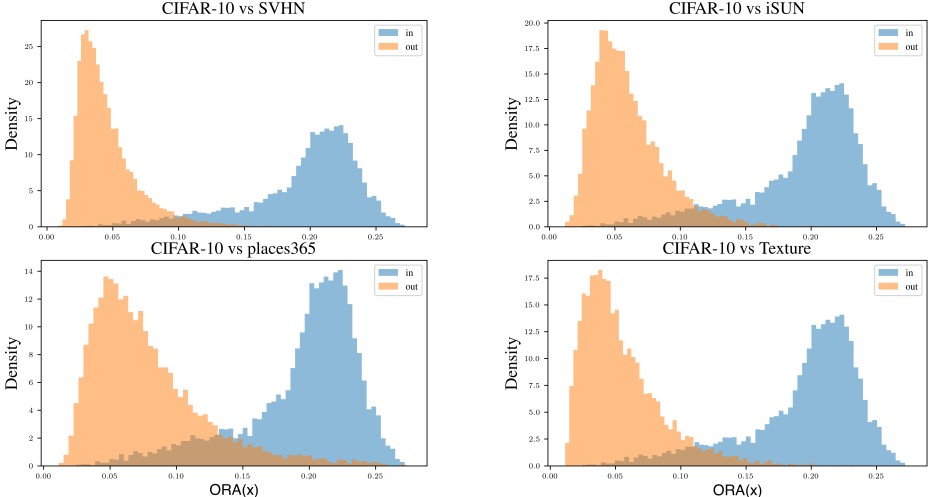

Figure 4: Score distributions of ID and OOD datasets in CIFAR-10 OOD Benchmark.

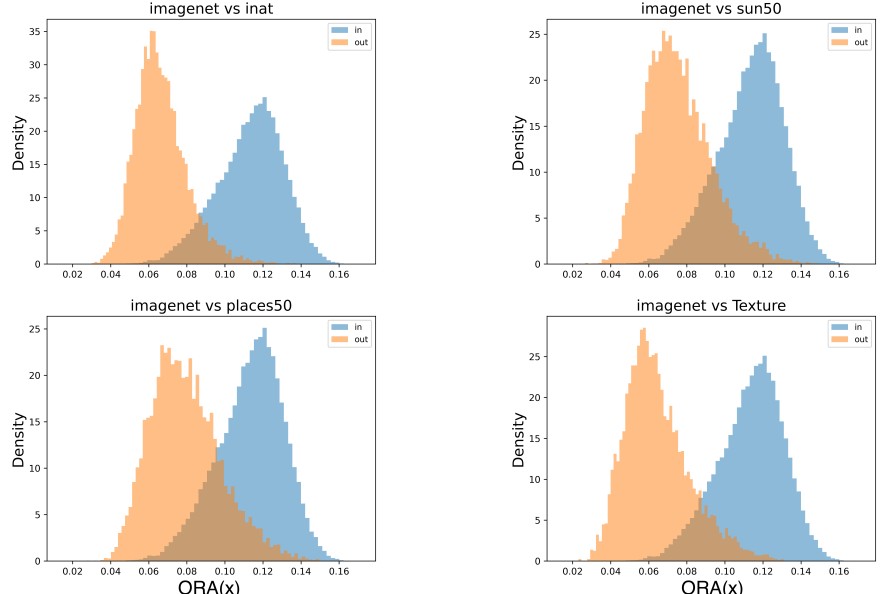

Figure 5: Score distributions of ID and OOD datasets in ImageNet OOD Benchmark.

# H Comparison of Different Angles

Figure 6 presents the score distributions for different angles discussed in Section 3.1. The results show that $\sin(\alpha)$ does not provide good ID/OOD separation, whereas $\sin(\theta)$ and $\sin(\theta)/\sin(\alpha)$ present significantly clearer distinctions. Additionally, incorporating $\sin(\alpha)$ into $\sin(\theta)/\sin(\alpha)$ hinders its performance compared to using $\sin(\theta)$ alone.

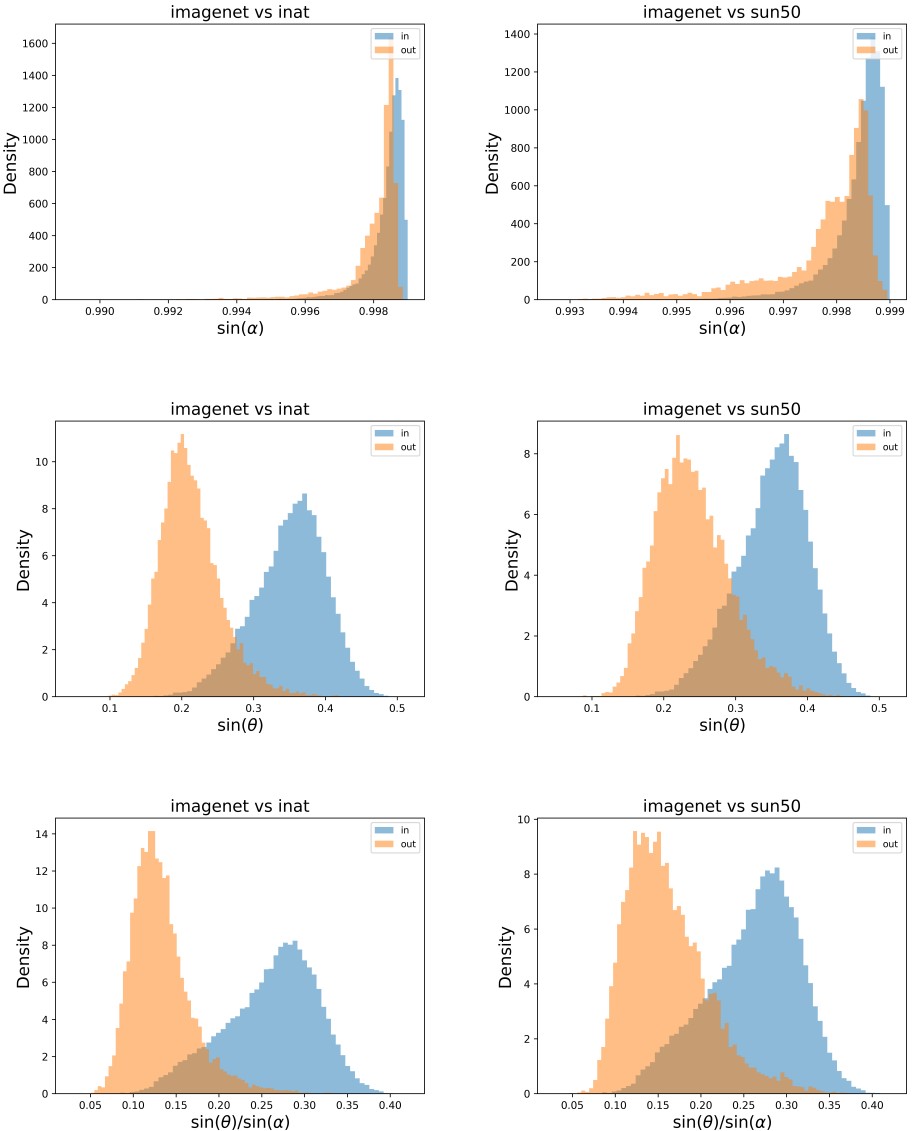

Figure 6: We demonstrate the ID/OOD separability of $\sin(\alpha)$, $\sin(\theta)$ and $\frac{\sin(\theta)}{\sin(\alpha)}$. Columns show the performances on iNaturalist and SUN datasets respectively. It can be seen that the ID/OOD class separability is the best when $\sin(\theta)$ is used: considering $\sin(\alpha)$ impedes the performance as confirmed quantitatively in terms of FPR95 and AUROC metrics in Table 9.

# I Near OOD Results

In Table 20, we report ORA's FPR and AUROC on NINCO [Bitterwolf et al., 2023] and SSB-hard [Vaze et al., 2022] datasets, along with average scores and in-distribution (ID) classification accuracy.

**NINCO dataset:** We observe that models pretrained on ImageNet21k generally outperform their non-pretrained counterparts in terms of FPR95, consistent with the observations in Bitterwolf et al. [2023]. However, no single method, including ORA, dominates across all architectures. For example, ORA attains the best results on the plain ConvNeXt Liu et al. [2022] backbone, while Rcos [Bitterwolf et al., 2023] outperforms others on DeiT [Touvron et al., 2021] and ResNet50 [He et al., 2016]. Interestingly, Energy [Liu et al., 2020] performs best on Swin-pretrained [Liu et al., 2021].

**SSB-Hard dataset:** A similar trend is evident: ORA outperforms other methods on plain ConvNeXt [Liu et al., 2022] and offers strong performance on SWIN-pre [Liu et al., 2021], ConvNeXt-pre [Liu et al., 2022], and DeiT-pre [Touvron et al., 2021], where it outperforms Rcos [Bitterwolf et al., 2023], RMaha [Ren et al., 2021], and Pcos [Galil et al.]. However, it is worse than naive baselines like Energy [Liu et al., 2020] in certain pretrained settings. In contrast, for models without pre-training, naive methods degrade significantly while ORA remains on par with the top baselines.

**Discussion**. These findings underscore a fundamental distinction between far- and near-OOD detection. Identifying completely novel inputs (as in far-OOD) is a fundamentally different problem from determining how much deviation is acceptable within or between known classes (as in near-OOD). Consequently, methods designed for detecting novelty or semantic outliers—such as those based on global uncertainty or energy scores—do not necessarily excel in near-OOD settings. This makes direct comparisons across these tasks potentially misleading. In near-OOD scenarios, where fine-grained class separation is critical, approaches that explicitly incorporate class-wise feature statistics—such as Rcos, Pcos, or RMah [Bitterwolf et al., 2023, Galil et al., Ren et al., 2021]—tend to have a clear advantage. These methods are better equipped to handle intra-class variance and subtle distributional shifts, highlighting the need for tailored evaluation and method design across OOD subtypes.

Table 20: OOD performance across benchmarks. FPR↓ and AUR↑ are reported for NINCO and SSB-Hard. The last two columns show the average performance and ID classification accuracy.

| Model | NINCO | | SSB-Hard | | Average | | ID Acc (%) |
|---|---|---|---|---|---|---|---|
| | FPR↓ | AUR↑ | FPR↓ | AUR↑ | FPR↓ | AUR↑ | |
| ResNet-50 | 68.31 | 85.89 | 85.65 | 71.17 | 76.98 | 78.53 | 76.12 |
| ResNet-50-SCL | 54.39 | 86.50 | 79.09 | 71.35 | 66.74 | 78.93 | 77.31 |
| ConvNeXt | 48.04 | 89.17 | 70.90 | 75.47 | 59.47 | 82.32 | 84.84 |
| ConvNeXt-pre | 45.54 | 90.56 | 68.82 | 77.13 | 57.18 | 83.84 | 85.49 |
| DeiT | 71.76 | 78.47 | 83.07 | 69.38 | 77.42 | 73.93 | 83.13 |
| DeiT-pre | 60.07 | 86.40 | 80.40 | 72.40 | 70.24 | 79.40 | 84.79 |
| Swin | 66.91 | 84.00 | 81.28 | 73.95 | 74.10 | 78.98 | 84.49 |
| Swin-pre | 52.62 | 88.25 | 77.45 | 74.76 | 65.04 | 81.51 | 85.73 |
| EVA | 46.61 | 88.15 | 67.86 | 75.65 | 57.24 | 81.90 | 87.88 |

## J  Relation to [Liu and Qin, 2024]

In this section we are going to describe the methodology relation of our method with respect to the method of Liu and Qin [2024], explaining the performance gap in favor of ORA.

Let $\mathbf{z} \in \mathbb{R}^D$ be a feature, $\mathbf{z}_{db}$ its orthogonal projection onto the decision boundary given in (1), and let $\boldsymbol{\mu}_{\text{ID}} \in \mathbb{R}^D$ denote the in-distribution mean. Define centered quantities

$$\tilde{\mathbf{z}} = \mathbf{z} - \boldsymbol{\mu}_{\text{ID}}, \qquad \tilde{\mathbf{z}}_{db} = \mathbf{z}_{db} - \boldsymbol{\mu}_{\text{ID}}.$$

In the triangle with vertices $\{\mathbf{0}, \tilde{\mathbf{z}}, \tilde{\mathbf{z}}_{db}\}$, the law of sines gives

$$\|\mathbf{z} - \mathbf{z}_{db}\| = \|\tilde{\mathbf{z}}\| \frac{\sin(\theta)}{\sin(\alpha)}, \tag{5}$$

where $\theta$ is the angle at $\tilde{\mathbf{z}}$ (discriminative ID/OOD signal, measured in the ID-centered frame) and $\alpha$ is the auxiliary angle at the origin $\mathbf{0}$; the radial term $\|\tilde{\mathbf{z}}\|$ captures nuisance scale (e.g., global brightness/contrast).

The fDBD score normalizes out the scale term:

$$\text{fDBD}(\mathbf{z}) = \frac{\|\mathbf{z} - \mathbf{z}_{db}\|}{\|\tilde{\mathbf{z}}\|} = \frac{\sin(\theta)}{\sin(\alpha)},$$

which makes the score invariant to uniform rescaling of features and improves over raw distance.

Empirically, $\sin(\alpha)$ is nearly identical for ID and OOD (see Fig. 2), adding variance without separation. ORA therefore retains only the informative angular component in the ID-centered frame:

$$s_{\text{ORA}}(\mathbf{z}) \propto \theta$$

**Hierarchy.**  Raw distance $d = \|\mathbf{z} - \mathbf{z}_{db}\|$ is confounded by the ID-centered norm $\|\tilde{\mathbf{z}}\| = \|\mathbf{z} - \boldsymbol{\mu}_{\text{ID}}\|$. The fDBD score removes this scale confound but still carries the factor $1/\sin\alpha$. ORA discards that factor and retains only the discriminative angle, which explains its per-model gains and its suitability for scale-invariant ensembling.

## K  Results on small datasets

To confirm empirically the hypothesis that ID points lie further from the decision boundary holds even for simple datasets and models, we trained a `LeNet-5` [LeCun et al., 2002] architecture on MNIST and measured the distance to the decision boundary for MNIST (ID) and Fashion-MNIST (OOD).

| Dataset | Mean | Std | Min | Max |
|---|---|---|---|---|
| MNIST | **5.86** | 0.62 | 2.69 | 7.49 |
| Fashion-MNIST | 4.77 | 0.82 | 1.99 | 6.66 |

Table 21: Distance to the decision boundary for MNIST (ID) and Fashion-MNIST (OOD).

ID samples are, on average, 1.1 units further from the decision boundary than OOD samples.

**Discussion.**  Cross-entropy training maximizes class-conditional log-likelihood, which drives features to align strongly with their corresponding weight vectors (confident regions). This places ID features deep inside their class cones and therefore further from the boundary. In contrast, OOD images do not match strong class-specific patterns, so their features align less and end up closer to the boundary, as reflected in the distance statistics.

This experiment supports empirically the assumption on the distribution of in-distribution points with respect to the decision boundaries which is at the core of our method and fDBD Liu and Qin [2024].

