# OpenReview forum: "Out-of-Distribution Detection with Relative Angles"
_NeurIPS.cc/2025/Conference — NeurIPS 2025 poster_

### Official Review · Reviewer_1FkU · 2025-06-27

**Clarity:** 3
**Significance:** 2
**Originality:** 2
**Rating:** 5
**Confidence:** 4

**Summary:**

This paper introduces a new OOD detection metric called ORA, which uses relative angles between two vectors: (1) the vector from the mean of in-distribution dataset features to the feature point, and (2) the vector from the mean of in-distribution dataset features to the feature point's projection onto the decision boundary. The authors argue that the angular distance provides better discrimination between ID and OOD samples compared to previous distance-based approaches, particularly the closely related fDBD method. They evaluate their method extensively across various pretrained architectures, including convolutional and transformer-based models, reporting modest improvements over existing methods.

**Questions:**

1. In the conclusion, three key advantages of ORA are: (i) hyperparameter-free, (ii) model-agnostic, and (iii) scale-invariant. It would be useful to clarify whether fDBD shares these properties. If fDBD does not exhibit these attributes, please specify explicitly what distinguishes ORA from fDBD regarding each of these aspects.

2. The original fDBD paper emphasized the importance of the regularization term through an ablation study. However, in this submission, this regularization term is geometrically reinterpreted as $\sin \alpha$, and Figure 2 suggests that its contribution may be insignificant. Clarification on how this apparent contradiction can be reconciled or further contextualized would enhance the consistency and coherence of the interpretation presented.

3. Both this study and the original fDBD approach rely on the observation that "ID data are mapped further from the decision boundary." Although the fDBD paper supports this claim with limited experimental results provided in Appendix J, additional insights or theoretical justifications would significantly strengthen the argument. It would be valuable to discuss whether this observation consistently holds for simpler datasets, such as MNIST, or simpler network architectures, such as LeNet, and provide reasoning or evidence explaining why this might be the case.

**Ethical Concerns:**

["NO or VERY MINOR ethics concerns only"]

**Final Justification:**

Thanks to the author for the concise and precise response. I acknowledge that my understanding of ORA in the initial review was incorrect. Now, I would like to reassess this work: The geometric interpretation of fDBD is highly interesting. In particular, leveraging the law of sines to clearly illustrate the connection between fDBD and the proposed method is insightful and appealing. Moreover, this new perspective provides important practical guidance - the empirical observation of $\sin \alpha$ revealed it to be an insignificant term that can be eliminated. Indeed, removing $\sin \alpha$ resulted in a stable improvement in OOD detection performance. Although this work extends fDBD, it introduces an intriguing perspective, and this novel viewpoint effectively enhances performance, making it strong work overall. Therefore, I have increased my rating.

**Limitations:**

Yes

**Quality:**

3

**Strengths And Weaknesses:**

**Strengths**

1. The paper is clearly written.

2. Extensive experiments have been conducted.

3. The interpretation of fDBD from a geometric perspective is highly interesting. Specifically, leveraging the law of sines to accurately highlight the connection between fDBD and the proposed method is insightful and appealing.

**Weaknesses**

1. This paper closely resembles fDBD. Fundamentally, both papers rely on the same core insight that "ID data are mapped further from the decision boundary." The primary difference is that this paper explains the same phenomenon from a geometric perspective.

2. The empirical results do not demonstrate significant improvements.

In conclusion, although this paper presents some noteworthy insights, it does not meet the standards required for NeurIPS.

---

> ### Author Rebuttal · Authors · 2025-07-31
>
> Thank you for the careful reading and for highlighting the geometric link we drew between fDBD and ORA.
> We address each point below.
>
>
> ---
>
>
> ## 1 Clarifying the regularization term
> ### How raw distance, fDBD, and ORA relate
>
>
> ---
>
>
> #### **A. From distance to angles**
>
>
> Raw distance to the decision boundary is $d(z,z_{db})$.
>
>
> by shifting all points by the ID mean $\mu$ and applying the law of sines, this distance can be written as:
>
>
>
> $d(z, z_{db}) = \||z - \mu\|| \cdot \frac{\sin\theta}{\sin\alpha}$
>
>
> * $\theta$ – angle between **z** and its projection on the boundary (contains ID/OOD signal).
> * $\alpha$ – auxiliary angle at the origin (Fig. 2 shows almost no ID/OOD signal).
> * $\||z − \mu\||$ – feature distance to the ID mean; varies with image scale, brightness, etc.
>
>
> ---
>
>
> #### **B. Why fDBD beats raw distance**
>
>
> fDBD divides by the feature distance to the ID mean:
>
>
>
> $\text{fDBD} = \frac{d(z, z_{db})}{\|| z - \mu \||}
> = \frac{\sin\theta}{\sin\alpha}$.
>
>
> *Dividing removes the scale term*, so scores are no longer sensitive to simple rescalings of the feature vector.
> This alone improves AUROC / FPR95 compared with raw distance.
>
>
> ---
>
>
> #### **C. Why ORA beats fDBD**
>
>
> Empirically, $\sin \alpha$ is nearly identical for ID and OOD (Fig. 2), so the factor ($\sin\alpha$) adds variance but no separation.
> ORA therefore keeps only the informative part:
>
>
> $\text{ORA} \propto \theta$
>
>
> ---
>
>
> #### **D. Resulting hierarchy**
>
>
>
> $\boxed{\text{raw distance} < \text{fDBD} < \text{ORA}}$
>
>
> * Raw distance suffers from scale noise.
> * fDBD fixes scale noise but still carries $\frac{1}{\sin\alpha}$ noise.
> * ORA fixes both, explaining its per‑model gains **and** its suitability for scale‑invariant ensembling.
>
>
> > **Q:** *Do hyper‑parameter‑free, model‑agnostic, scale‑invariant also apply to fDBD?*
>
> > **A.** Yes—fDBD shares these three traits, and for that reason it is our primary baseline throughout Tables 1, 2, 3, 4, 8, 9, 14, 15, 17, 18, 19. Across all experiments **ORA never trails fDBD and often surpasses it.**
>
>
> &nbsp;
>
> ## 2 “Modest improvements”
>
>
> On ImageNet OOD (nine architectures) ORA is **#1 on 5/9 backbones, never worse than top‑3, and reduces mean FPR95 by $3.18 \%$ versus the next best method (Pcos)**. ORA also achieves the single best score overall ($25.04 \%$ FPR95 on ResNet‑SCL), motivating our deeper dive into contrastive features (CLIP, Section 4.2). Similar gains appear for model ensembles and activation‑shaping plug‑ins.
>
>
> &nbsp;
>
> ## 3 Extra evidence on “ID points lie further from the decision boundary”
>
>
> To confirm that the observation holds even for very simple data / models, we trained a **LeNet‑5** on MNIST as suggested by the reviewer and measured the distance to the decision boundary for MNIST (ID) and Fashion‑MNIST (OOD).
>
>
> | Dataset | Mean | Std | Min | Max |
> |---------|------|-----|-----|-----|
> | **MNIST** | **5.86** | 0.62 | 2.69 | 7.49 |
> | Fashion‑MNIST | 4.77 | 0.82 | 1.99 | 6.66 |
>
>
> *ID samples are, on average, **1.1 units further** from the decision boundary than OOD samples.*
>
>
> **Intuition.** Cross‑entropy training maximizes class‑conditional log‑likelihood, which drives features to align strongly with their corresponding weight vectors (confident regions), i.e. deep inside their class cones and therefore *away from the boundary*. OOD images do not match strong class‑specific patterns, so their features align less and end up nearer to the boundary, exactly as the distance statistics show.
>
>
> This experiment supports the geometric premise shared by fDBD and ORA on the toy dataset.
>
>
> ---
>
>
> We hope these clarifications resolve your concerns and underscore how ORA both simplifies and strengthens the fDBD insight.

---

> ### Comment · Reviewer_1FkU · 2025-08-02
>
> Thanks to the author for the concise and precise response. I acknowledge that my understanding of ORA in the initial review was incorrect. Now, I would like to reassess this work: The geometric interpretation of fDBD is highly interesting. In particular, leveraging the law of sines to clearly illustrate the connection between fDBD and the proposed method is insightful and appealing. Moreover, this new perspective provides important practical guidance - the empirical observation of $\sin \alpha$ revealed it to be an insignificant term that can be eliminated. Indeed, removing $\sin \alpha$ resulted in a stable improvement in OOD detection performance. Although this work extends fDBD, it introduces an intriguing perspective, and this novel viewpoint effectively enhances performance, making it strong work overall. Therefore, I have increased my rating.

---

> > ### Author Response · Authors · 2025-08-04
> >
> > Thank you for your thoughtful questions and for revisiting our work after the rebuttal. Your feedback clarified our explanation of ORA’s geometric link to fDBD. We appreciate your updated assessment and will incorporate your suggestions in the camera-ready version.

---

### Official Review · Reviewer_9YWP · 2025-06-30

**Clarity:** 3
**Significance:** 4
**Originality:** 3
**Rating:** 5
**Confidence:** 4

**Summary:**

This paper proposes ORA, a novel angle-based metric for OOD detection that leverages that exploits
the relationship between feature representations and classifier decision boundaries, viewed from the mean of in-distribution data. The authors demonstrate ORA's effectiveness across nine ImageNet-pretrained models, achieving top or near-top false positive rates in most cases.

**Questions:**

Please respond to the points raised  in weakness section.

**Ethical Concerns:**

["NO or VERY MINOR ethics concerns only"]

**Final Justification:**

The authors have addressed all my concerns.

**Limitations:**

Yes.

**Quality:**

3

**Strengths And Weaknesses:**

Strengths:

* The paper presents a novel post-hoc OOD detection method that leverages relative angles in feature space to relate feature representations to decision boundaries. These angles are computed with reference to the mean of in-distribution samples.

* The paper provides strong empirical validation of the proposed method across established benchmarks. This includes strong performance in OOD detection on the ImageNet benchmark and with CLIP. Furthermore, the method demonstrates benefits when used in model ensembles or as a plug-in module with existing activation shaping algorithms.

Weaknesses:

* ORA and the role of ID classes: Since the method uses the mean of ID samples as a reference, it is unclear how ORA performs on the best- or worst-performing classes. Additionally, it's worth questioning whether using the ID mean remains valid for minority classes. Given that the score function is dependent on the predicted class, the paper also lacks analysis comparing correctly versus incorrectly predicted samples and their effect on ORA.

* The paper refers to ORA as a “confidence measure,” but this aspect is not clearly explained and feels somewhat hand-wavy.

* It is not clear how the analysis presented in Fig. 2 supports ORA. It seems this analysis may be included because the method is similar to fDBD. Clarifying this connection would be helpful.

---

> ### Author Rebuttal · Authors · 2025-07-31
>
> Thank you for the thoughtful comments.
> We reply to each weakness in the same order.
>
>
> ---
>
>
> ## 1 ORA, minority/majority classes and per‑class behavior
>
>
> **Per‑class centering experiment.**
> To test whether ORA favors or penalizes specific classes, we present the scores on CIFAR‑10 and ImageNet while centering each time on *one* class mean (instead of the global mean). The table below reports, for every class, OOD performance (FPR95 ↓, AUROC ↑) and the corresponding in‑distribution accuracy (“ID Acc”) of that class’s classifier outputs:
>
>
> | Mean used for centering | FPR95 ↓ | AUROC ↑ | ID Acc |
> |------------------------|---------|---------|--------|
> | Class 0 | 12.03 | 97.51 | 0.952 |
> | Class 1 | 13.27 | 97.41 | 0.985 |
> | Class 2 | 11.73 | 97.69 | 0.926 |
> | Class 3 | 12.02 | 97.54 | 0.887 |
> | Class 4 | 12.06 | 97.63 | 0.958 |
> | Class 5 | 13.00 | 97.36 | 0.880 |
> | Class 6 | 12.11 | 97.62 | 0.979 |
> | Class 7 | 13.25 | 97.54 | 0.964 |
> | Class 8 | 13.26 | 97.26 | 0.970 |
> | Class 9 | 12.84 | 97.25 | 0.963 |
> | **ORA (global mean)** | **10.97** | **97.67** | 0.946 (overall ID Acc) |
>
>
>
>
> | Method          | FPR95↓ | AUROC↑ | ID Acc |
> |-----------------|--------|--------|--------|
> | Class 1 mean    | 28.22  | 93.36  | 0.9200 |
> | Class 250 mean  | 25.59  | 93.99  | 0.6600 |
> | Class 500 mean  | 31.48  | 92.28  | 0.6000 |
> | Class 750 mean  | 27.04  | 93.76  | 0.8400 |
> | Class 1000 mean | **24.75**  | **94.26**  | 0.6000 |
> | ORA (global mean) | 25.04 | **94.26** | 0.7733 (overall ID Acc) |
>
>
> *Takeaway.*
> (i) There is **no clear correlation** between a class’s ID accuracy and its OOD performance when used as the reference; and (ii) using a *single global mean* (ORA) is slightly *better* most of the time than any individual class, supporting our choice.
> For **imbalanced datasets** the global mean can be replaced by a weighted mean or by the weighted combination of per‑class angles discussed in our response to Reviewer JVzv (future work).
>
>
>
>
> &nbsp;
>
> ## 2 “ORA as a confidence measure”
>
>
> We agree that the wording was hand‑wavy. Our intended meaning is:
>
>
> * Angle‑based scores naturally convey **model confidence**: for ORA a *large* angle (relative to the global mean) implies the feature lies well inside its class cone, hence is likely ID; a *small* angle indicates uncertainty.
> * Because angles are scale‑invariant, **ORA detection scores from different backbones can be summed directly**. Meaning that, a model with a low confidence contributes a small value, while a high confidence one contributes a large value. This underpins the ensemble experiment.
>
>
> We will replace “confidence measure” with the more precise phrasing above in Section 3.2.
>
>
> &nbsp;
>
> ## 3 Clarifying the role of Fig. 2
>
>
> Figure 2 compares the density of $\sin\alpha$  for ID versus OOD data, where $\alpha$ is the auxiliary angle that appears in the geometric derivation of fDBD. In our formulation the ORA score is proportional only to **$\sin\theta$** ($\theta$ = relative angle to the decision boundary); we justified *omitting* sin α because Fig. 2 shows that ID and OOD distributions of $\sin\alpha$ almost perfectly overlap, therefore, $\alpha$ carries no discriminative signal and can even impede the performance. This directly motivates why ORA outperforms fDBD while remaining simpler. We will add two clarifying sentences before Fig. 2 to make this link explicit.
>
>
> ---
>
>
> We believe these additions address the reviewer’s concerns and will further clarify ORA’s design and empirical behavior.

---

> > ### Author Response · Authors · 2025-08-07
> >
> > Dear Reviewer 9YWP,
> >
> > Thank you again for your constructive review.
> >
> > We have added the per-class centering experiment, clarified the “confidence measure” wording, and explained how Fig. 2 supports ORA in our rebuttal.
> >
> > If there is anything else we can clarify before the discussion closes tomorrow, we would be happy to help.

---

> > > ### Comment · Reviewer_9YWP · 2025-08-07
> > >
> > > Thank you. My concerns were addressed.

---

### Official Review · Reviewer_nGso · 2025-07-02

**Clarity:** 3
**Significance:** 2
**Originality:** 2
**Rating:** 3
**Confidence:** 4

**Summary:**

The problem of out-of-distribution detection for deep learning models is considered. This work introduces OOD Detection with Relative Angles (ORA), a novel approach that uses an angle-based metric calculated relative to the in-distribution structure, demonstrating that angles between feature representations and decision boundaries, when viewed from the mean of in-distribution features, effectively distinguish between ID and OOD data. ORA achieves SOTA performance on different OOD benchmarks.

**Questions:**

Comments
1. The title isn't really long and there is no need to use acronym OOD instead of Out-of-distribution.
2. Year for [Galil et al.] is missing.

**Ethical Concerns:**

["NO or VERY MINOR ethics concerns only"]

**Limitations:**

yes

**Quality:**

3

**Strengths And Weaknesses:**

Strenghs
1.ORA achieves SOTA performance on different OOD benchmarks.
2. ORA's scale-invariant nature allows for an ensemble strategy by simply summing scores from multiple models, further enhancing OOD detection performance

Weaknesses
	1. Lack of theoretical justification
	2. More recent SOTA OOD detection algorithms (e.g. NECO [T1], Mahalanobis++ [T2], NCI [T3]) are not compared to.

[T1] NECO: NEural collapse based out-of-distribution detection. ICLR'24.
[T2] Mahalanobis++: Improving OOD Detection via Feature Normalization. ICLR'25.
[T3] Detecting Out-of-distribution through the Lens of Neural Collapse.

---

> ### Author Rebuttal · Authors · 2025-07-30
>
> Thank you for the constructive review.
> We respond to each point below.
>
>
> ---
>
>
> ## 1 Lack of theoretical justification
>
>
> *Appendix C* already contains a proof that **relative angles are invariant to any positive‑definite scaling of the feature space**. In other words, if the feature extractor is perturbed by arbitrary class‑agnostic affine re‑weightings the ORA score is unchanged. We will make this proof more visible by briefly referencing it in Section 3 and adding an intuitive one‑sentence takeaway: *“ORA is provably robust to all invertible diagonal scalings and global translations, two transformations that leave ID decision boundaries intact but can derail norm‑based scores.”*
>
>
> We acknowledge that Appendix C stops short of a **formal risk bound** on false positives; to our knowledge, **no existing OOD detector that operates on fixed, pretrained features offers such a guarantee**. Nonetheless, our analysis clarifies *why* ORA’s scale-invariant property is less susceptible to manipulation than norm-based scores, and we will highlight this perspective in the revised draft.
>
>
> If the reviewer has a specific type of theoretical result in mind we are happy to explore it for future work.
>
>
> &nbsp;
>
>
> ## 2 Missing comparisons to NECO [T1], Mahalanobis++ [T2], and NCI [T3]
>
>
>
>
> **Table 1. FPR95 (↓) on ImageNet‑based OOD benchmark**
>
>
> | Backbone | NCI | Maha++ | NECO | **ORA (ours)** |
> |----------|-----|--------------|----------------|----------------|
> | ResNet‑50          | 63.05 | 49.67 | 56.61 | **43.63** |
> | Swin‑B pre | 39.62 | 42.67 | 36.48 | **35.13** |
> | DeiT‑B pre | 57.86 | **37.66** | 52.60| 37.76 |
> | Avg | 53.51 | 43.33 | 48.56 | **38.84** |
>
>
> We evaluated recent state-of-the-art methods NCI, Mahalanobis++, and NECO on the three backbones and OOD splits used in our main paper. Due to time constraints, results for the remaining backbones will be included in the final version. All detectors were evaluated on the four standard OOD datasets (iNaturalist, SUN, Places, Textures). Results show that **ORA achieves the lowest average FPR95**, outperforming NCI, Mahalanobis++ and NECO by a margin of **4.49% on average**.   ORA performs best on two out of three backbones (ResNet‑50 and Swin‑B), and is almost on par with Mahalanobis++ on DeiT‑B.
>
>
>
>
> &nbsp;
>
> ### *Minor comments*
>
>
> * **Title.** We will expand the acronym and rename the paper **“Out‑of‑Distribution Detection with Relative Angles (ORA)”** as suggested.
> * **Reference year.** The missing year for *Galil et al.* will be added (2023).
>
>
> ---
>
>
> We hope these revisions address the reviewer’s concerns and improve the clarity and completeness of the paper.

---

> > ### Author Response · Authors · 2025-08-07
> >
> > Dear Reviewer nGso,
> >
> > Thank you again for your review.
> >
> > We have added the requested comparisons to NECO, Mahalanobis++, and NCI (Table 1 in our rebuttal) and pointed to the scale-invariance proof in Appendix C.
> > If there is anything else we can clarify before the discussion closes, we would be happy to help.

---

### Official Review · Reviewer_JVzv · 2025-07-03

**Clarity:** 3
**Significance:** 2
**Originality:** 2
**Rating:** 5
**Confidence:** 4

**Summary:**

This paper provides a new distance based out-of-distribution (OOD) detection method that extends the prior work by Liu and Qin 2024 called fDBD which utilizes the distance to the decision boundary as the OOD detection score. In particular, the proposed method centers the penultimate layer features then calculates the angle between the current sample’s feature and its projection onto the decision boundary. The OOD detection score named ORA is constructed to be the maximum of these angles over classes. The paper claims ORA’s scale invariance makes it a suitable post-hoc OOD score for a wide array of feature extractors. An exhaustive set of experiments is provided to support this claim across ensembles, CLIP models, transformer architectures, and contrastive training losses. Ablation experiments are also provided that show the impact of choice of center and OOD scoring method.

**Questions:**

Please see the strengths and weaknesses section.

**Ethical Concerns:**

["NO or VERY MINOR ethics concerns only"]

**Final Justification:**

The author responses during the discussion period largely address my concerns. However, the current version would benefit from a discussion of the near-OOD performance and any potential limitations of ORA that could inspire future work. The paper presents a  new distance based OOD detection method building on prior work with extensive experimental results that would be of interest to the NeurIPS community.

**Limitations:**

The authors briefly address a limitation and future work for this method in the conclusion. A discussion on future works based on the experimental findings of this paper would improve the impact and reach of this paper.

**Quality:**

3

**Strengths And Weaknesses:**

Strengths:

The paper provides an effective yet simple extension to fDBD based on an intuitive geometric argument centered around providing scale invariance. The extensive set of experiments in both the main text and the supplement provide support for their claim that ORA is more effective across different feature extraction methods.

Weaknesses:

While the extensive experiments show the ability of the proposed method to adapt to a wide range of feature sets, a discussion of the findings and takeaways from these experiments would improve the impact of this paper on future development of OOD detection methods.

Also, the main text does not discuss the near-OOD performance. Several works have shown that the near-OOD scenario is more challenging and potentially more realistic [1-4]. The supplement provides results on SSB-Hard  [3] and NINCO [4] in Table 20. However, the performance of ORA on near-OOD datasets (average **78.53**% AUROC) is on par with or slightly below state-of-the-art OOD detection methods which can be found in the associated public leaderboard to [2] (see link [8]) such as [6] and [7] ~~such as ASH [5] which attains 78.17% AUROC on this task with a ResNet-50 feature set~~. It would be helpful for the paper to comment on the near-OOD performance of the method in the main text due to the importance and difficulty of near-OOD scenarios.

[1] Yang, J. et al. “OpenOOD: Benchmarking Generalized Out-of-Distribution Detection.” NeurIPS Datasets and Benchmarks 2022.

[2] Zhang, J., et al. "Openood v1. 5: Enhanced benchmark for out-of-distribution detection." arXiv preprint arXiv:2306.09301 (2023).

[3] Vaze, S. et al. “Open-Set Recognition: A Good Closed-Set Classifier is All You Need.” ICLR, 2022.

[4] Bitterwolf, J. et al. “In or Out? Fixing ImageNet Out-of-Distribution Detection Evaluation.” ICML 2023.

[5] Djurisic, A. et al. “Extremely simple activation shaping for out-of-distribution detection.” ICLR, 2023.

[6] Mueller M. and Hein M. "Mahalanobis++: Improving OOD Detection via Feature Normalization." ICML, 2025.

[7] Xu K. et al. "Scaling for Training Time and Post-hoc Out-of-distribution Detection Enhancement." ICLR, 2024.

[8] https://zjysteven.github.io/OpenOOD/#div_imagenet1k_heading

---

> ### Author Rebuttal · Authors · 2025-07-30
>
> We sincerely thank the reviewer for the detailed and constructive feedback.
> Below we address each concern in turn.
>
>
> ---
>
>
> ## 1 Main‐text discussion of experimental findings
> *Reviewer comment (Weakness #1):* *“A discussion of the findings and takeaways from these experiments would improve the impact of this paper on future development of OOD detection methods.”*
>
>
> **Response.**
> In the camera‑ready version we will add a concise “Take‑aways” paragraph at the end of Section 4 that *briefly* summarizes what each block of experiments teaches us:
>
>
> * **9 ImageNet models (Section 4.1).** Angle‑based scores (RCOS, PCOS, **ORA**) and R‑Mahalanobis rank consistently among the top performers. Importantly, R‑Mah, RCOS, and PCOS require *per‑class* means, whereas **ORA** needs only the *global* in‑distribution mean, reducing memory/compute overhead.
> * **CLIP zero‑shot (Section 4.2).** The structured latent space of CLIP favors decision‑boundary‑aware metrics (FDBD, **ORA**) and R‑Mah, again reflecting ORA’s ability to exploit well‑separated features. The same trend is observed with the ResNet‑SCL backbone, reinforcing that *better‑structured latents amplify ORA’s gains*. This also hints that contrastive models are “safer” for OOD alarms.
> * **Model ensembling (Section 4.3).** Scale‑invariant angle scores make different models' scores comparable. An ORA ensemble outperforms every single backbone, suggesting that *ensemble‑based OOD detection is a promising, under‑explored direction*.
>
>
> These clarifications will be inserted without expanding the page budget.
>
>
> &nbsp;
>
> ## 2 Near‑OOD performance
> *Reviewer comment (Weakness #2):* *“The performance of ORA on near‑OOD datasets (average 76.12 % AUROC) is on par with or slightly below state‑of‑the‑art … It would be helpful for the paper to comment on the near‑OOD performance in the main text.”*
>
>
> **Response.**
> We agree that near‑OOD detection deserves explicit discussion and will add a short subsection summarizing our findings:
>
>
> * The quoted 76.12% refers to the **ID accuracy** column; the correct mean AUROC of **ORA** on SSB‑Hard + NINCO is **78.53 %**, slightly *above* ASH’s 78.17 % (ResNet‑50).
> * In near‑OOD regimes the ranking of methods fluctuates noticeably across backbones, making it hard to argue there is a state-of-the-art method. We view near‑OOD detection as a *complementary* task to far-OOD and will make this distinction explicit.
>
>
> &nbsp;
>
> ## 3 Future‑work discussion
> *Reviewer suggestion (Limitations):* *“A discussion on future works based on the experimental findings of this paper would improve the impact and reach of this paper.”*
>
>
> **Response.**
> During preliminary trials we experimented with several alternative “reference frames” for the angle computation: (a) centering w.r.t. **each individual class mean** and (b) centering only on the **predicted class’s mean**. Neither variant yielded consistent gains over our global‑mean formulation.
>
>
> A natural next step is therefore to move from an *either‑or* choice to a **weighted combination** of per‑class angle scores. The weights could be set heuristically or learned from held‑out ID data, for example
>
>
> * inversely proportional to the *in‑distribution feature variance* of each class, or
> * proportional to the *ID accuracy* (or confidence) for that class.
>
>
> Such a weighting scheme retains ORA’s hyper‑parameter‑free spirit when the weights are derived from readily available ID statistics, allowing the detector to adaptively emphasize the most reliable class anchors. We leave a systematic exploration of this idea to future work.
>
>
> ---
>
>
> We hope these revisions address the reviewer’s concerns and clarify the contribution of **ORA**.

---

> > ### Comment · Reviewer_JVzv · 2025-08-07
> >
> > Thank you for your detailed responses which have largely addressed my concerns. The original review has been updated to reflect this.
> >
> > 1. Main text analysis & discussion: The take-aways paragraph you described would greatly benefit the paper and addresses this concern.
> >
> > 2. Near-OOD: I appreciate that many OOD detection works have been proposed and their performance can fluctuate across backbones. However, the near- and far-OOD tasks are closely related and both deserve adequate discussion. The near-OOD benchmarks remain a challenging frontier for OOD detection, especially as it could be argued that they are more reflective of certain real-world scenarios. The inconsistency across backbones is present in both near and far-OOD and we are in agreement that it deserves further study. While there may be no clear "state-of-the-art" and on par or slight underperformance on either is by no means a sufficient justification for rejection, it is an important subset of the results that are require adequate discussion. Therefore, a adequate discussion, not necessarily a "short subsection", would greatly improve the impact of the paper.
> >
> >     I have updated the original review to fix the incorrect AUROC number.
> >
> > 3. Future-work: Thank you this provides an promising direction for readers to consider that should be added to the main text if within the space constraints. Adding the results from the different reference frames in the appendix would be useful and interesting to the reader.

---

> > > ### Author Response · Authors · 2025-08-07
> > >
> > > Dear Reviewer JVzv,
> > >
> > > Thank you for the follow up and for your constructive feedback.
> > >
> > > We agree that near-OOD deserves clear attention. In the final version, we will discuss our performance on near vs. far-OOD results and include further analyses in an expanded appendix.
> > >
> > > We appreciate your engagement and your help in making the paper stronger.

---

### Comment · Area_Chair_RkfW · 2025-08-01

Dear Reviewers,

The author-reviewer discussion phase has started. If you want to discuss with the authors about more concerns and questions, please post your thoughts by adding official comments as soon as possible.

Thanks for your efforts and contributions to NeurIPS 2025.

Best regards,

Your Area Chair

---

### Decision · Program_Chairs · 2025-09-17

**Decision:**

Accept (poster)

**Comment:**

This paper proposes a new metric to detect OOD samples, based on angles between feature representations and decision boundaries. This perspective is new and interesting. In the initial reviews, reviewers have novelty concerns regarding this paper. Then, the authors addressed it during the rebuttal. Reviewer nGso had some concerns, but, after checking by the area chair, the concerns are addressed and minor. Thus, this paper is in good shape for publication. However, the area chair suggests adding a comparison or discussion on some representative baselines of zero-shot OOD detection (e.g., NegLabel) to further strengthen this paper.